# Detecting critical nodes in forest landscape networks to reduce wildfire spread

**Denys Yemshanov**[1]*, **Ning Liu**[1], **Daniel K. Thompson**[1], **Marc-André Parisien**[2], **Quinn E. Barber**[2], **Frank H. Koch**[3], **Jonathan Reimer**[4]

**1** Natural Resources Canada, Canadian Forest Service, Great Lakes Forestry Centre, Sault Ste. Marie, Ontario, Canada, **2** Natural Resources Canada, Canadian Forest Service, Northern Forestry Centre, Edmonton, Alberta, Canada, **3** USDA Forest Service, Southern Research Station, Eastern Forest Environmental Threat Assessment Center, Research Triangle Park, North Carolina, United States of America, **4** Capital Regional District, Victoria, British Columbia, United Kingdom

* denys.yemshanov@canada.ca

**Data Availability Statement:** All relevant data are within the manuscript and its Supporting information files.

## Abstract

Although wildfires are an important ecological process in forested regions worldwide, they can cause significant economic damage and frequently create widespread health impacts. We propose a network optimization approach to plan wildfire fuel treatments that minimize the risk of fire spread in forested landscapes under an upper bound for total treated area. We used simulation modeling to estimate the probability of fire spread between pairs of forest sites and formulated a modified Critical Node Detection (CND) model that uses these estimated probabilities to find a pattern of fuel reduction treatments that minimizes the likely spread of fires across a landscape. We also present a problem formulation that includes control of the size and spatial contiguity of fuel treatments. We demonstrate the approach with a case study in Kootenay National Park, British Columbia, Canada, where we investigated prescribed burn options for reducing the risk of wildfire spread in the park area. Our results provide new insights into cost-effective planning to mitigate wildfire risk in forest landscapes. The approach should be applicable to other ecosystems with frequent wildfires.

## Introduction

Wildfires, while being a natural ecosystem process in many biomes, can pose significant economic and social threat to human communities in forested regions [1–3]. Land management agencies invest significant resources into the prevention and suppression of wildfires in forest landscapes and yet the economic costs associated with fires continue to increase rapidly [4], amounting to as much as $348B annually in the United States alone [3]. In part, the escalating costs are driven by the extreme challenges of managing wildfires in rugged landscapes, such as in western North America, where drought and complex terrain combine to trigger catastrophic fires. In these landscapes, where wildfires may endanger human lives, deploying fire response resources can be prohibitively expensive or sometimes logistically impossible.

Preventive fuel treatments, such as prescribed burns or strategic thinning of forest stands, are intended to decrease the probability of fire spread and reduce fire severity and,

**Funding:** Funding for this work was provided by Natural Resources Canada, Canadian Forest Service Wildfire Research Management Program.

**Competing interests:** The authors have declared that no competing interests exist.

consequently, the damage to human infrastructure. While there is a consensus that a substantial reduction of flammable biomass will reduce fire spread and severity, factors such as the location, size, maintenance, and use in fire operations may undermine fuel-treatment effectiveness [5,6]. If effective, fuel treatments can reduce the costs of fire suppression activities, and potentially save human lives [7–9]. Effective implementation of fuel treatments can result in smaller average fire sizes, as well as reduced occurrence of large fires, which, in turn, can lead to suppression cost savings [10]. Yet, fuel treatments can be difficult to plan effectively in complex (e.g., mountainous) landscapes [11–14]. In fact, the overall effectiveness of fuel reduction programs has met with some skepticism, largely due to the suboptimal placement of fuel treatments [15,16]. Limited resources and personnel, as well as imperfect understanding of fire behaviour, necessitate careful planning of fuel treatments to maximize their effectiveness.

Optimization has been widely used to support decisions about fire prevention and suppression [17–20]. Several linear programming models have been proposed to assist with the planning of wildfire prevention and fuel treatments aimed to reduce the severity of future fires in the landscape and their potential spread [20–27]. The proposed models featured an objective of fragmenting the landscape to minimize fire hazard (i.e., the combination of fire spread and intensity) [22] and were formulated as single- or multi-period site treatment problems with variable fuel accumulation rates [19,23]. Kabli et al. [28] proposed a two-stage stochastic integer programming method to allocate fuel treatment to minimize the total treatment cost and expected future losses. The optimal fuel treatment models of Minas et al. [19] and Rachmawati et al. [27] considered a landscape with multiple land-cover types and estimated fuel loads as a function of vegetation age. Acuna et al. [29] and Alonso-Ayuso et al. [30] proposed a harvest planning model incorporating the creation of fire breaks to minimize wildfire risk while also achieving a desired harvesting objective. Optimal design of prescribed burns was proposed in Alcasena et al. [31] and Matsypura et al. [24]. Rytwinski and Crowe [32] applied a stochastic simulation-optimization approach to evaluate the performance of fire-breaks allocations generated by a metaheuristic. Konoshima et al. [33,34] integrated a fire simulation model into a two-period stochastic dynamic model to find spatial allocations of timber harvest and fuel management in the face of spatially endogenous fire risk. Their approach used a fire simulation model to enumerate all possible fire occurrence patterns in all plausible treatment decisions and considered the trade-offs between fire risk, timber harvest value and fuel treatment cost. Similarly, the optimal fuel treatment model of Wei [21] minimized the total expected loss from fires using fire spread predictions made with a fire simulation model. The loss from fire was calculated as the sum of losses in all locations within the fire perimeter weighted by the probability of fire ignition and by the probability of a fire lasting for a given period after ignition.

Several of the proposed models minimized connectivity between forest patches with high wildfire risk as a way to reduce fire spread potential in a landscape [27]. Minas and Hearne [35] outlined a fuel treatment model that clustered sites where prescribed burns were scheduled. León et al. [25] described a model that tracked connectivity between adjacent sites representing candidates for prescribed burn actions. Pais et al. [14] proposed a maximum-weight connected subgraph problem to allocate harvesting to reduce the spread of fires. Matsypura et al. [24] applied a network optimization approach for the planning of prescribed burns. Their model depicted a flammable landscape as a network of connected patches (nodes), where each node was characterized by a fuel load value. They adopted a critical node detection problem (CND) [36–39] that, similar to other models, set an objective to minimize connectivity between nodes with high fuel loads. However, the potential spread of fires was considered only between adjacent nodes with fuel loads above a chosen threshold. This assumption enabled a simplified formulation of the fuel treatment problem by activating/deactivating nodes with high fuel loads in a particular planning period, but it also made the solutions

sensitive to the chosen fuel load threshold value. Notably, a node-based fuel accumulation or fire hazard metric does not fully characterize the potential of fires to spread across a landscape (i.e., from node to node). Ideally, a risk metric should quantify directional spread of fires from one location to another. Such a metric provides a better ecological foundation for composing a landscape-level fire spread network because it depicts relevant factors, such as prevailing winds during periods of elevated fire spread potential, that are known to influence landscape-level fire spread patterns.

In this paper, we employ a directional metric that quantifies the probability of fires spreading between a pair of locations in a landscape to solve a fuel treatment problem: allocating a set of prescribed burns to minimize the chance that wildfires will spread through the landscape, subject to an upper bound on the total treated area of prescribed burning or similar fuel reduction treatments (henceforth referred to as "prescribed burning"). For each pair of forest patches, we estimate the probability that a fire ignited in one patch will spread to another. We incorporate this metric into a modified Critical Node Detection (CND) problem [36–39], which we apply to solve a prescribed burn planning problem. We demonstrate the approach with a case study in Kootenay National Park, British Columbia, Canada, where we examine prescribed burn options aimed at reducing the risk of fires in the park.

## Materials and methods

### Detecting critical nodes in a network of flammable forest sites

A forest landscape can be thought of as a connected network of flammable patches (nodes), where the connecting arcs (edges) depict possible vectors of fire spread between adjacent patches. To minimize the possibility of fires spreading widely across the area, the manager allocates a set of treatments (prescribed burns) among the nodes. Treating a node helps reduce fire intensity [40] to the point of more effective suppression [41]. For simplicity, we assume that the treatment of a node is equivalent to removing a node and all arcs connected to that node from the forest landscape network, although we acknowledge that fuel treatments are not always completely effective [42]. The manager's problem is determining the best way to partition the network so that the probability of fires spreading through the area is minimized.

A popular strategy for solving this problem is to reduce the connectivity between nodes with flammable fuels in the landscape network. This strategy can be implemented by solving a CND problem, which finds the key nodes in a network whose removal maximally degrades the connectivity of the network according to a chosen metric [36–38,43–45]. The concept of critical nodes characterizes the vulnerability of a network after a portion of nodes are removed, which, depending on the type of network, could be the result of natural disasters or technical failures. The CND problem has been applied in many disciplines [46] including security applications [47,48], transportation [49], social network analysis [50] and epidemiology [51–54].

Let $G = (N,E)$ be a graph with a set of $N$ nodes (vertices) and a set of edges $E$, $E \subset N \times N$ (Fig 1a). We assume that two nodes $i$ and $j$ are connected if the graph $G$ contains a path between them. We define a connected component of graph $G$ as a subgraph in which each node has a path to every other node in the component but not to any node outside that component. A graph $G$ is fully connected and represents a single connected component if all its nodes are pairwise connected. If there is a pair of nodes that are not connected, then graph $G$ contains at least two connected components (Fig 1b).

We define a binary variable $x_i \in \{0,1\}$, $i \in N$, where $x_i = 1$ if node $i$ is *not* deleted from graph $G$ and $x_i = 0$ otherwise. A binary variable, $u_{ij} \in \{0,1\}$, is defined for every pair of nodes $i,j \in N$ and assumes that $u_{ij} = 1$ if both nodes $i$ and $j$ are *not* deleted (i.e., $x_i = x_j = 1$) and there is a path connecting $i$ and $j$, i.e., when nodes $i$ and $j$ are in the same connected component (Fig 1c). In

a)  b)  c)

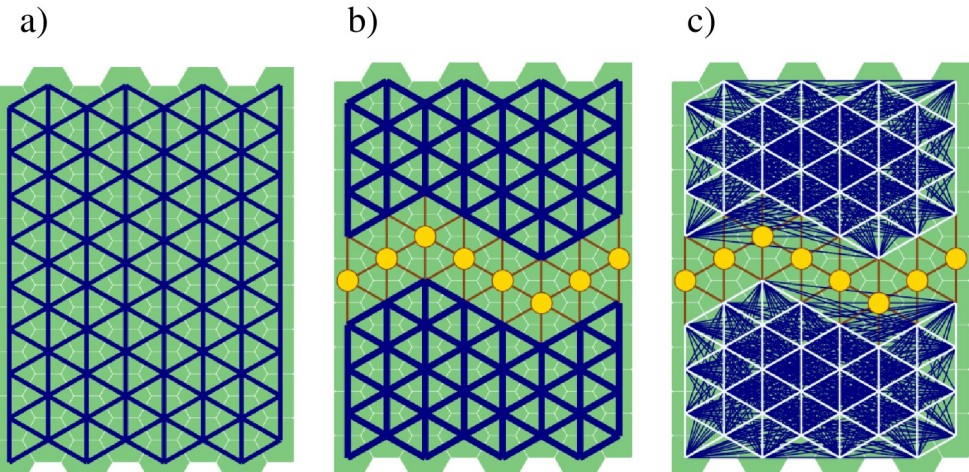

**Fig 1.** a) A landscape network of connected patches (graph) $G$ as a single connected component before interdiction; b) An interdicted graph with two connected components. Large dots denote the removed nodes; c) Pairwise connections $u_{ij}$ between nodes in the connected components after interdiction (lines in dark blue).

our fire prevention context, the existence of a path between nodes $i$ and $j$ means a fire could spread between $i$ and $j$. The area of a node $i$ is $c_i$. The subset of nodes that can be removed from a graph $G$ is limited by an upper bound treatment area limit $B$ (which also defines the total node removal cost). The graph obtained after a removal of $R$ critical nodes is a subgraph (s) of $G$ composed of the set of remaining nodes, $N \setminus R$ (Fig 1b and 1c).

A common and, in our case, highly relevant objective for the CND problem is to minimize the total number of connected node pairs in the remaining components (subgraphs) after removal of $R$ critical nodes [36,37]. Below, we provide the CND problem formulation based on [37], i.e.:

$$\min \sum_{\substack{i<j \\ i,j \in N}} u_{ij} \tag{1}$$

s.t.:

$$\sum^{i \in N} c_i (1 - x_i) \leq B \tag{2}$$

$$u_{ij} \geq x_i + x_j - 1 \quad \forall \quad (i,j) \in E \tag{3}$$

$$u_{ij} \geq \frac{1}{M} \sum_{\substack{k \neq j}}^{k \in N_G(i)} u_{kj} - (1 - x_i) \quad \forall \quad i,j \in N, \quad i \neq j, \quad (i,j) \notin E, \tag{4}$$

$x_i, u_{ij} \in \{0,1\} \ \forall \ i,j \in N, i \neq j$.

The formulation (1–4) yields $O(|N|^2)$ constraints and is more efficient than the original CND problem formulation with triangular inequalities [36], which yields $O(|N|^3)$ constraints. In our fire prevention context, objective function (1) minimizes the total number of node pairs $ij$ with possible spread of fires between $i$ and $j$ in area $N$. Constraint (2) sets an upper bound on the total treatment area. Constraint (3) ensures, for adjacent nodes $i$ and $j$, that $u_{ij} = 1$ if neither

node is deleted. Constraint (4) ensures that nodes $i$ and $j$ are connected if there is a non-deleted node $k$, $k \neq j$, in the connected component $\aleph_G(i)$ that includes $i$ (i.e., the neighbourhood of $i$), as well as nodes with non-interdicted path connections to $i$, such that $k$ and $j$ are connected.

## A modified CND problem for fire prevention planning

For each node $i$, the CND problem (1–4) evaluates the presence of path connections to all other nodes $j$ in network $N$, $i \neq j$ [37,39] (Fig 2a). In our case, the presence of a path connection between nodes $i$ and $j$ allows a fire ignited in $i$ to spread to $j$. In real landscapes, a fire ignited in a particular location could potentially spread over some area around the ignition point but is unlikely to spread across the entire landscape (Fig 2b). The fire extent can be limited by natural barriers, unavailability of fuel, fire suppression or the number of days with weather conditions conducive to fire spread prior to heavy rain or snowfall [55–61]. Thus, for a given node $i$, one only must evaluate the path connections to those $j$ nodes to which a fire ignited in $i$ realistically could spread (Fig 2b, shaded area). This greatly reduces the number of node pairs $ij$, decision variables $u_{ij}$ and constraint inequalities (3) and (4) in the CND problem.

Our approach of building a fire spread network $G$ differs from the previous CND problem application in Matsypura et al. [24], where a fire spread network included all nodes with fuel loads above a chosen threshold. Reliable estimation of that threshold is difficult because a forest landscape is characterized by a continuum of flammable fuel loads [55,62], and fire ignitions and spread can be driven by factors other than local fuel amounts, such as topography and variable weather [63,64]. Furthermore, a node-based fuel load value does not represent the factors controlling the directional spread of fires, such as synoptic weather patterns [65] that influence topographic wind funnelling [66].

We build network $G$ from the probabilities of fire spread between pairs of locations (nodes) in landscape $N$. For each node $i$, we define a subset, $\Omega_i$, of nodes $j$ that are potential spread destinations of fires ignited in $i$. Collectively, the destination nodes $j$ in subset $\Omega_i$ delineate the possible spread extent for a fire ignited at $i$ (Fig 2b and 2c). The idea is analogous to a *fireshed* concept [67] (Fig 2d), which defines the locations $j$ around a location of interest $i$ that could be the source of a fire that spreads to $i$. In our case, however, we define the fireshed from the opposite direction, i.e., as the area around node $i$ to which fires ignited in $i$ could potentially spread (Fig 2b).

Defining the subsets $\Omega_i$ helps incorporate available spatial information on potential fire spread into the CND problem. For each node $i$, we restrict the set of decision variables $u_{ij}$ and constraints (3) and (4) to the nodes $j$ that are in subset $\Omega_i$. We also weight the values of the binary decision variable $u_{ij}$ in objective function (1) by the probabilities of fire spread from node $i$ to node $j$, $p_{ij}$. Weighting $u_{ij}$ by these probabilities prioritizes interdiction of the likeliest fire spread paths. We then formulate a modified CND model (problem 1 hereafter) as follows:

$$\min \sum_{\substack{i \in N \\ j \neq i}}^{i \in N} \sum_{j \neq i}^{j \in \Omega_i} u_{ij} p_{ij} \tag{5}$$

s.t.: constraint (2) and

$$u_{ij} \geq \frac{1}{M} \sum_{\substack{k \neq j}}^{k \in N_G(i)} u_{kj} - (1 - x_i) \quad \forall \quad i \in N, j \in \Omega_i, \quad i \neq j, \quad (i,j) \notin E \tag{6}$$

$$u_{ij} \geq x_i + x_j - 1 \quad \forall \quad (i,j) \in E, j \in \Omega_i, \tag{7}$$

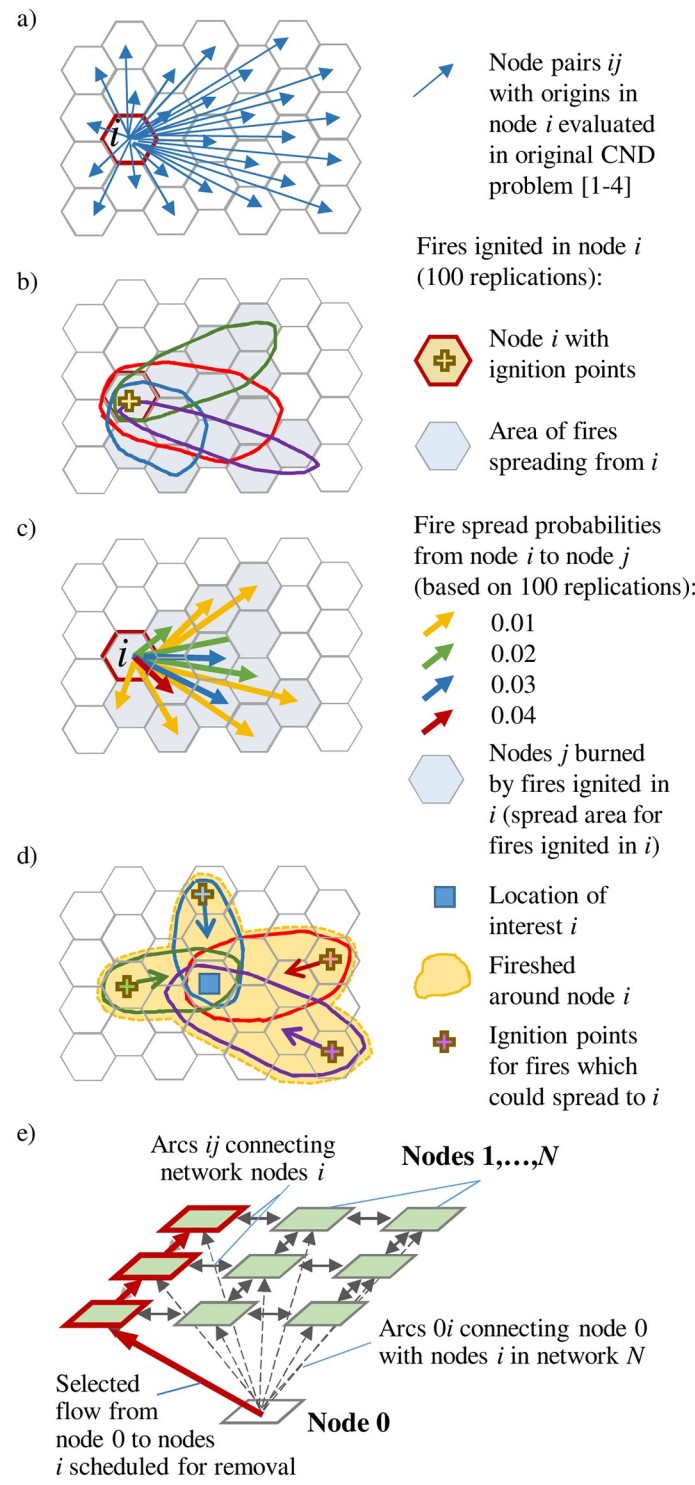

**Fig 2.** a) Node pairs *ij* with the origin in node *i* that are evaluated in the basic CND problem; b) potential spread area of fires ignited in location *i*. Nodes with the positive probabilities of a fire spread from the ignition node *i* are shaded; c) probabilities of a fire spread from node *i* with the ignition points to nodes *j*, $p_{ij}$. Fire spread examples in Fig 2a and 2b are based on 100 iterations; d) a fireshed around the node of interest *j*; e) using auxiliary node 0 to inject the flow into the landscape network. Dashed arrows show the arcs connecting adjacent nodes *N*. Arrows in bold red show the flow from node 0 through the connected nodes scheduled for removal (outlined in red).

$x_i, u_{ij} \in \{0,1\} \; \forall \; i \in N, i \in \Omega_i, i \neq j.$

Objective function (5) minimizes the expected number of node pairs between which fire spread is possible after the removal of some nodes in network *G*. When nodes have equal area, the upper bound *B* limits the total number of removed nodes. As already noted, the subset $\Omega_i$ includes those *j* nodes that are potential destinations of fires ignited in node *i*, with fire spread probabilities $p_{ij} > 0$. Constraints (6) and (7) are comparable to (3) and (4) but consider only node pairs with non-zero fire spread probabilities. For each pair of nodes *i* and *j*, objective (5) tracks the path connections $u_{ij}$ and $u_{ji}$ and fire spread probabilities $p_{ij}$ and $p_{ji}$ in both directions, which helps account for differences in the directional spread of fires. For example, the probability values $p_{ij} > p_{ji}$ could characterize the spread of fires between nodes *i* and *j* with prevailing winds blowing from *i* to *j*.

Objective (5) can be applied to evaluate two distinct scenarios. A *probabilistic fireshed* scenario uses the unaltered spread probability values $p_{ij}$ to scale the decision variables $u_{ij}$ in objective (5). In this scenario, the node pairs *ij* exhibiting the highest fire spread frequencies have the greatest impact on the objective value. These frequencies depend on the size distribution of fires ignited in node *i* and spreading to *j* because fires ignited in *i* must have the size at least as large as the distance between *i* and *j*. In a forest landscape, the fire size distribution follows a general power law function [58,59,68–73]. Consequently, this scenario tends to interdict by means of geographically distinct hotspots with small but frequent fires across the landscape.

In a second *binary fireshed* scenario, the $p_{ij}$ values in objective function (5) are replaced with binary fire spread indicators, $p_{ij\,\text{bin}}$, that are equal to one for $p_{ij} > 0$ and zero for $p_{ij} = 0$. Thus, the scenario assumes equal unary weights for all node pair connections with $p_{ij} > 0$. For each node *i*, the sum of the $p_{ij\,\text{bin}}$ values defines the possible spread extent of fires ignited in *i*, a fireshed area that the scenario minimizes. For a typical gridded network of forest sites, the number of possible connections between node pairs grows in quadratic proportion to the linear size of a connected component (and so the distance between a pair of nodes *i* and *j*). Accordingly, the binary fireshed scenario minimizes the number of long paths between node pairs and therefore minimizes the spread of large fires.

The probabilistic and binary fireshed scenarios can be viewed as alternative strategies for minimizing potential fire spread. The trade-off between these strategies can be examined by replacing the $p_{ij}$ (or $p_{ij\,\text{bin}}$) values in objective Eq (5) with their weighted average, i.e.: $p_{ij}(1 - \varphi) + p_{ij\,\text{bin}}\varphi$, where $\varphi$ is the scaling factor. The trade-off frontier can be found by solving the problem for a range of $\varphi$ values between 0 and 1.

## Controlling the spatial contiguity of the prescribed burns

By itself, the CND problem does not control for connectivity between the removed nodes. The removal of multiple nodes may occur in clusters that cover a substantial area. In our fire prevention context, removal of a node indicates implementation of prescribed burning in the node area. Often, the maximum size of a prescribed burn must be constrained for safety reasons. Prescribed burns require careful planning and costly logistical support to ensure safe execution [74]. High costs and logistical challenges limit the total number of prescribed burns that can be executed in an area over a season [75]. For the sake of efficiency, individual burns must be spatially contiguous, but their size cannot exceed the safety limit. We address these requirements by adding a network flow sub-problem for controlling the number and spatial contiguity of the subsets of removed nodes (burns) in the CND model. We add an auxiliary node 0 to the set of landscape nodes *N*. Node 0 is connected to all nodes *i* in *N* by arcs 0*i* and serves as the source of a flow that can only be passed to nodes *i* scheduled for removal (i.e.,

with $x_i = 0$) (Fig 2e). A node scheduled for removal can receive the flow from node 0 (or another node with $x_i = 0$). It retains one unit of flow and passes the flow to an adjacent node scheduled for removal, and so on until all nodes scheduled for removal receive flow. The total number of connected nodes that can be removed is limited by the burn size limit $A$. For each planned burn (planning step hereafter), only one connection between node 0 and nodes $i$ is allowed, thus creating one connected subset of removed nodes (a single burn).

We introduce a set $T$ to define the total number of contiguous subsets of removed nodes (i.e., planned burns) in landscape $N$. Each planning step $t$, $t \in T$, allocates one subset of removed nodes with the maximum size limit $A$. For each arc $ij$ connecting adjacent nodes $i$ and $j$, $i,j \in \{0\}, E$, a binary variable $z_{ijt}$ selects the flow through arc $ij$ in step $t$ (so that $z_{ijt} = 1$ when arc $ij$ is selected and $z_{ijt}$ otherwise). A non-negative decision variable $y_{ijt}$ defines the amount of flow via arc $ij$ between nodes $i$ and $j$ in step $t$. The updated CND formulation for planning of $T$ contiguous burns (problem 2 hereafter) is formulated as follows:

Objective (5)

s.t.: constraints (2,6,7) and

$$A_{\min} \leq \sum_{i=1}^{N} y_{0it} \leq A \; \forall \; t \in T \tag{8}$$

$$\sum_{j=0}^{\Theta_i} y_{jit} - \sum_{k=1}^{\Theta_i^+} y_{ikt} = \sum_{j=0}^{\Theta_i} z_{jit} \quad \forall \quad i \in \{0\}, N, t \in T, \quad (j,i) \in \{0\}, E, (i,k) \in E \tag{9}$$

$$y_{ijt} \leq z_{ijt} M \quad \forall \quad i \in \{0\}, N, j \in N, \quad (i,j) \in \{0\}, E \tag{10}$$

$$z_{ijt} \leq y_{ijt} \quad \forall \quad i \in \{0\}, N, j \in N, \quad (i,j) \in \{0\}, E \tag{11}$$

$$\sum_{t=1}^{T} \sum_{j=0}^{\Theta_i} z_{jit} \leq 1 \quad \forall \quad i \in N \tag{12}$$

$$\sum_{i=1}^{N} z_{0it} = 1 \quad \forall \quad t \in T \tag{13}$$

$$x_i = 1 - \sum_{t=1}^{T} \sum_{j=0}^{\Theta_i} z_{jit} \quad \forall \quad i \in N. \tag{14}$$

Constraint (8) ensures that the number of nodes that can receive flow, and the corresponding size of the removed node set $t$, stays within the range [$A_{\min}$; $A$]. Constraints (9–12) ensure that the nodes removed in step $t$ are connected in one segment. Constraint (9) ensures that the amount of incoming flow to node $i$ is equal to the amount of outgoing flow from $i$ to adjacent nodes plus the retained capacity of $i$ (one unit of flow). Set $\Theta_i$ denotes adjacent nodes $j$ (including node 0) that can pass flow to node $i$, while set $\Theta_i^+$ denotes adjacent nodes $k$ that can receive flow from $i$. Constraint (10) ensures no flow if arc $ij$ is not selected and constraint (11) specifies no selection of arc $ij$ if no flow occurs between nodes $i$ and $j$ in step $t$. Constraint (12) ensures that the flow to node $i$ from other nodes (including node 0) over $T$ steps comes through no more than one arc. This guarantees no overlap between the node selections in different steps $t$. Constraint (13) specifies a single connection from node 0 to nodes $i$ in step $t$, which yields one

contiguous subset of nodes. Constraint (14) ensures agreement between the node removal variable $x_i$ in the CND sub-problem and the selection of flow between adjacent nodes scheduled for removal. The term $\sum_{j=0}^{\Theta_i} z_{jit}$ defines the number of arcs with positive flow to node $i$, $i \in N$, and is equal to one if node $i$ receives flow from one of its adjacent nodes $j$ (or node 0) and zero otherwise. Table 1 lists the model parameters and variables.

Problem 2 assumes a single-period planning cycle where the CND sub-problem is solved once for $T$ planned burns. The time to find a feasible solution can be reduced by replacing objective (5) and constraint (13) with a penalty formulation, i.e.:

$$\min \sum_{i \in N} \sum_{\substack{j \neq i}}^{i \in \Omega_i} u_{ij} p_{ij} + \sum_{t=1}^{T} V_t f \qquad (15)$$

**Table 1. Summary of the model variables and parameters.**

| Symbol | Parameter/variable name | Description |
|---|---|---|
| *Sets*: | | |
| $N$ | Nodes (forest patches) $i,j$ in a landscape network (graph) $G$ | $i,j \in N$ |
| $E$ | Edges connecting adjacent nodes in a landscape network $G$ | $E \subset N \times N$ |
| $\aleph_G(i)$ | Connected component which includes node $i$ | $\aleph_G(i) \in N$ |
| $\Omega_i$ | Nodes $j$–potential spread destinations of fires ignited in node $i$ | $\Omega_i \in N$ |
| $\Theta_i^+$ | Adjacent nodes $k$ which can receive flow from node $i$ | $\Theta_i^+ \in N$ |
| $\Theta_i$ | Adjacent nodes $j$ (or node 0) which can pass flow to node $i$ | $\Theta_i \in N$ |
| $T$ | Planning steps $t$ in problem 2 | $t \in T$ |
| $T'$ | Planning time periods $t'$ in problem 3 | $t' \in T'$ |
| *Decision variables*: | | |
| $x_i$ | Node deletion binary variable ($x_i = 0$ if node is removed and $x_i = 1$ otherwise) | $x_i \in \{0,1\}$ |
| $u_{ij}$ | Binary variable defining that nodes $i$ and $j$ are *not* removed and there is a path connecting $i$ and $j$ | $u_{ij} \in \{0,1\}$ |
| $z_{ijt}$ | Binary flow indicator between nodes $i$ and $j$ which are scheduled for removal in step $t$ | $z_{ijt} \in \{0,1\}$ |
| $y_{ijt}$ | Flow through an arc $ij$ between adjacent nodes $i$ and $j$ scheduled for removal in step $t$ | $y_{ijt} \geq 0$ |
| $u_{ijt''}$ | Binary variable defining that nodes $i$ and $j$ are *not* removed and there is a path connecting $i$ and $j$ in period $t'$ | $u_{ijt''} \in \{0,1\}$ |
| $x_{it''}$ | Node removal binary variable ($x_{it'} = 0$ if node is removed in planning period $t'$ and $x_{it'} = 1$ otherwise) | $x_{it''} \in \{0,1\}$ |
| $V_t$ | Penalty for the number of connections from an auxiliary node 0 to the nodes scheduled for removal in step $t$ above one (creates a single contiguous set of removed nodes in step $t$) | $V_t \geq 0$ |
| *Parameters* | | |
| $p_{ij}$ | Fire spread probability from node $i$ to node $j$ | $p_{ij} \in [0;1]$ |
| $p_{ij\,bin}$ | Fire spread probability binary indicator: $p_{ij\,bin} = 1$ for $p_{ij} > 0$ and $p_{ij\,bin} = 0$ otherwise | $p_{ij\,bin} \in \{0,1\}$ |
| $w_{ij}$ | Fie spread probability between adjacent nodes $i$ and $j$, $ij \in E$ | $w_{ij} \in [0;1]$ |
| $c_i$ | Removal cost for node $i$ | $c_i = 1$ |
| $B$ | Node removal (treatment) area limit | $B > 0$ |
| $A_{min}$ | Minimum size of a contiguous set of the removed nodes in step $t$ | $A_{min} = 2$ |
| $A$ | Maximum size of a contiguous set of the removed nodes in step $t$ | $A = 6$ |
| $M$ | Large positive value | $M > 0$ |
| $f$ | Scaling factor for penalty $V_t$ | $f \geq 0$ |
| $\varphi$ | Scaling factor between the $p_{ij}$ values and their binary indicators $p_{ij\,bin}$ | $\varphi \in [0;1]$ |

s.t.: constraints (2,6–12,14) and

$$V_t \geq \sum_{i=1}^{N} z_{0it} - 1 \quad \forall \quad t \in T. \tag{16}$$

Objective (15) is similar to (5) except it includes a penalty, $V_t$, for each step $t$, adjusted by the scaling factor $f$. Constraint (16) defines the $V_t$ value as the number of connections from node 0 to nodes $i$ in area $N$ in step $t$ exceeding one. Setting a large scaling factor $f$ for $V_t$ instructs the model to schedule one contiguous set of nodes for removal for each step $t$.

Problem 2 solves a time-invariant CND problem for $T$ planned burns. Site treatments are typically implemented in a stepwise fashion over a defined time period. Our problem 3 accounts for the cumulative nature of multi-step planning; namely, the actions taken in the first step have the most impact on the system and may thus affect the actions taken in the sub-sequent steps. Problem 3 extends problem 2 to a multi-period case where the CND problem is solved for each period $t'$ and the cumulative impact of gradual node removals is tracked over the full timespan $T'$. The treatments with the greatest impact on fire spread are allocated first, followed by less impactful treatments. For each period $t'$, we define binary decision variables $u'_{ijt}$ and $x'_{ij}$ similarly to the binary variables $u_{ij}$ and $x_i$ in the time-invariant problems 1 and 2, and formulate the multi-period node removal problem as follows:

$$\min \frac{1}{T'} \sum_{t'=1}^{T'} \sum^{i \in N} \sum_{j \neq i}^{j \in \Omega_i} u'_{ijt'} p_{ij} + \sum_{t'=1}^{T'} V_{t'} f \tag{17}$$

subject to constraints (8–12,16) and

$$x'_{it'} = 1 - \sum_{v=1}^{t'} \sum_{j=0}^{\Theta_i} z_{jiv} \quad \forall \, i \in N, t' \in T' \tag{18}$$

$$u'_{ijt'} \geq \frac{1}{M} \sum_{k \neq j}^{k \in N_G(i)} u'_{kjt'} - \left(1 - x'_{it'}\right) \forall i \in N, j \in \Omega_i, i \neq j, (i,j) \notin E, t' \in T' \tag{19}$$

$$u'_{ijt'} \geq x'_{it'} + x'_{jt'} - 1 \quad \forall \quad (i,j) \in E, j \in \Omega_i, t' \in T'. \tag{20}$$

Objective (17) is similar to (15) but minimizes the expected number of node pairs with possible fire spread after the removal of some nodes in each period $t'$, over $T'$ periods. Given an anticipated short planning horizon in our study, we used a single set of fire spread probabilities $p_{ij}$ over timespan $T'$ but acknowledge that longer-duration periods $t'$ may require defining time-dependent sets of $p_{ijt'}$ values for each period $t'$.

As specified by constraint (18), tracking the connections between node pairs in the CND sub-problem (17,19,20) in period $t'$ includes all nodes removed over a timespan of $1, \ldots, t'$ periods, representing a portion of the full timespan $T'$. Subscript $v$ in (18) is an alias to subscript $t'$, $v \in T'$. The summation of flow variables $z_{ijv}$ over periods $1, \ldots, t'$ in (18) estimates whether node $i$ has received flow during time span $[1; t']$. Recall that a node $i$ scheduled for removal in period $t'$ must receive flow, so $\sum_{j=0}^{\Theta_i} z_{jit'} = 1$. Given that constraint (12) limits the flow to node $i$ over $T'$ periods to one instance only, the right side of Eq (18) can be either zero, when $i$ has received flow over time span $[1;t']$, or one otherwise. Thus, for each period $t'$, the CND sub-

problem tracks the impact of cumulative disruption of the landscape network $G$ over timespan $[1; t']$. Constraints (19) and (20) are analogous to (6) and (7) but applied at each period $t'$.

The key difference between problems 2 and 3 is that problem 3 solves the CND sub-problem for each period $t'$ to account for a cumulative reduction of fire spread potential after each period. This makes problem 3 combinatorially harder than problem 2. To reduce the solving time, we used the problem 2 solutions to initialize problem 3. We first solved problem 3 with constraint (21) that forces the node removal pattern to conform to the problem 2 solution (assuming the same scenario assumptions and the treatment area limit $B$), i.e.:

$$x'_{it'} \geq \chi_i \quad \forall \quad i \in N, t' \in T', \tag{21}$$

where $\chi_i$ denotes the values of decision variable $x_i$ in problem 2 solution. Under constraint (21), the problem 3 model finds an optimal timing for $T$ burns scheduled in the problem 2 solution. This solution was then used to warm start the original problem 3 model (8–12,16–20).

### Calculating the fire spread probabilities $p_{ij}$

For all node pairs $i,j$ in the landscape, the CND model requires knowledge of the probability that a fire ignited in $i$ will spread to $j$, $p_{ij}$. It is unnecessary to approximate the particular spread path(s) from $i$ to $j$ because the spread probability value $p_{ij}$ only defines the likelihood that a fire which is ignited in location $i$ will spread to location $j$ and does not require specification of how the fire might spread from $i$ to $j$. Evaluating the presence of a path (i.e., any path) connecting nodes $i$ and $j$ is handled by the CND model constraints (3) and (4).

We used a spatial fire simulation model to estimate the fire spread probabilities $p_{ij}$ between node pairs $ij$ in landscape $N$. Fire simulation models generate stochastic ignition events and plausible perimeters of fires spreading from the ignition locations [60–62,76,77]. Perimeters of individual fires are generated by combining ignitions and weather with a spatial fire growth model. Fire simulation models are popular tools for mapping potential wildfire spread and, as such, estimate the likelihood of wildfires [55,60,62,78–87]. Recent computational advances permit simulation of ignitions, fuel distribution, impacts of terrain and weather, and generate many plausible fire spread patterns [86]. Fire simulation models have been used to evaluate fuel treatment projects [55,88] and evaluate fire suppression strategies [77,88]. Examples of popular fire models include the Canadian Burn-P3 model [78], the FSim model in the USA [80], and the Australian Phoenix model [89].

We used the Burn-P3 model [78] to generate stochastic fire ignitions and spread scenarios in our study area (see Supplement S1). Burn-P3 implements the crown fire scheme of the Canadian Fire Behaviour Prediction System (FBP) [90], modelling surface fires as well as the transition to crown fires (and the rate of crown fire spread itself). For each iteration, Burn-P3 generates the ignition locations and perimeters of individual fires. We calculated the fire spread probabilities $p_{ij}$ from these outputs as follows. First, we generated a hexagonal network of interconnected forest patches containing flammable fuels (nodes $N$ in a fire spread network $G$ in the CND problem). For a hexagon (node) $i$, we selected all fires ignited in $i$ and overlaid their perimeters on network $G$ (see Supplement S2 Fig 1a in S2 File). For each fire ignited in $i$, we selected all nodes $j$ to which that fire was able to spread (S2 Fig 1b in S2 File). Each selected node pair $ij$ was assigned a value of 1 (S2 Fig 1c and 1d in S2 File), which denoted a fire spread event from $i$ to $j$. This procedure was sufficient to count the number of fire spread events from $i$ to $j$; approximation of the actual spread paths for the events was unnecessary. We repeated this calculation for all fires generated by Burn-P3. For every node pair $ij$, we summed the number of times a fire ignited in $i$ spread to $j$ and divided this sum by the number of Burn-P3

iterations (S2 Fig 1d in S2 File). This yielded the probability of a fire ignited in node *i* spreading to node *j*, $p_{ij}$, which we used in the probabilistic fireshed scenarios. Converting all $p_{ij} > 0$ to one gave us the binary fire spread indicators $p_{ij\ \text{bin}}$, which we used in the binary fireshed scenarios.

## Case study

We applied the CND model in the Kootenay National Park, British Columbia, Canada. The park is part of a UNESCO World Heritage Site in the Canadian Rocky Mountains (Fig 3) and covers a subalpine and alpine region with elevations between 1200 and 3400 m above sea level (Fig 4a) [88]. Forests at lower elevations in the area are dominated by Engelmann spruce (*Picea engelmannii*), subalpine fir (*Abies lasiocarpa*), white spruce (*Picea glauca*), and lodge-pole pine (*Pinus contorta*) and are replaced by alpine tundra at upper elevations [91]. Lightning ignitions are a common cause of large stand-replacing fires [92]. Small, lower-intensity

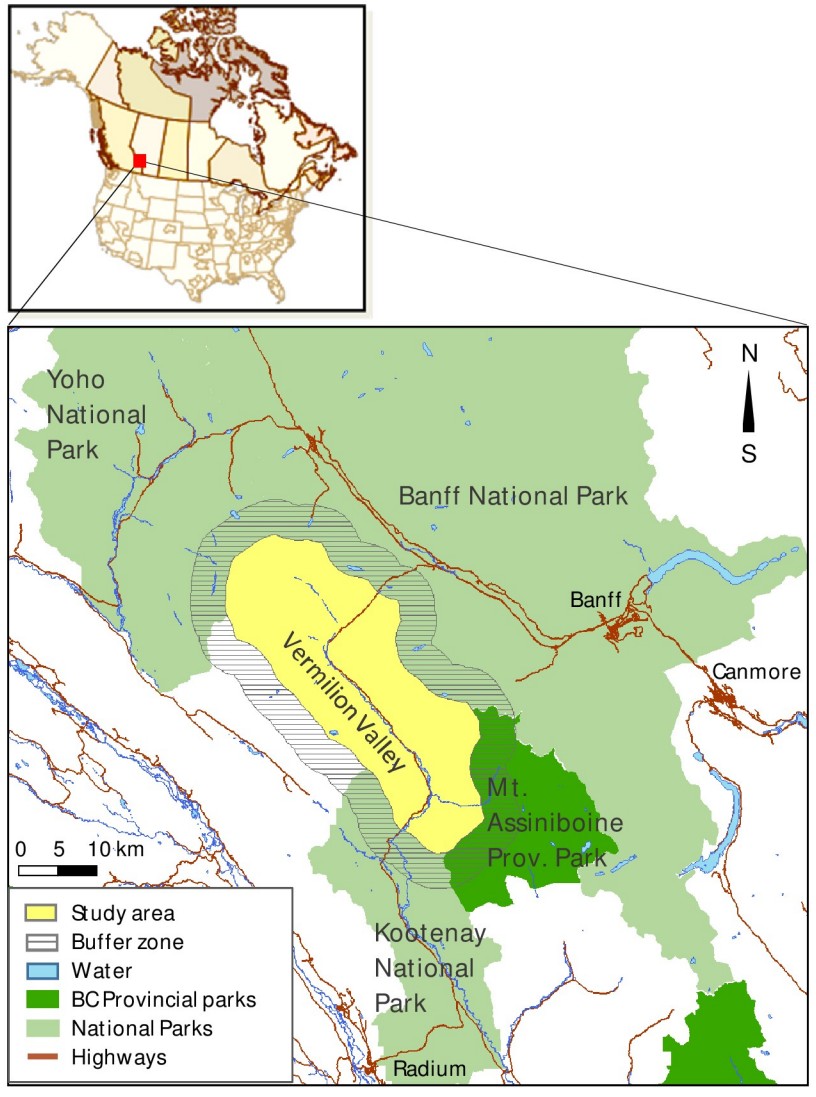

**Fig 3. Study area.**

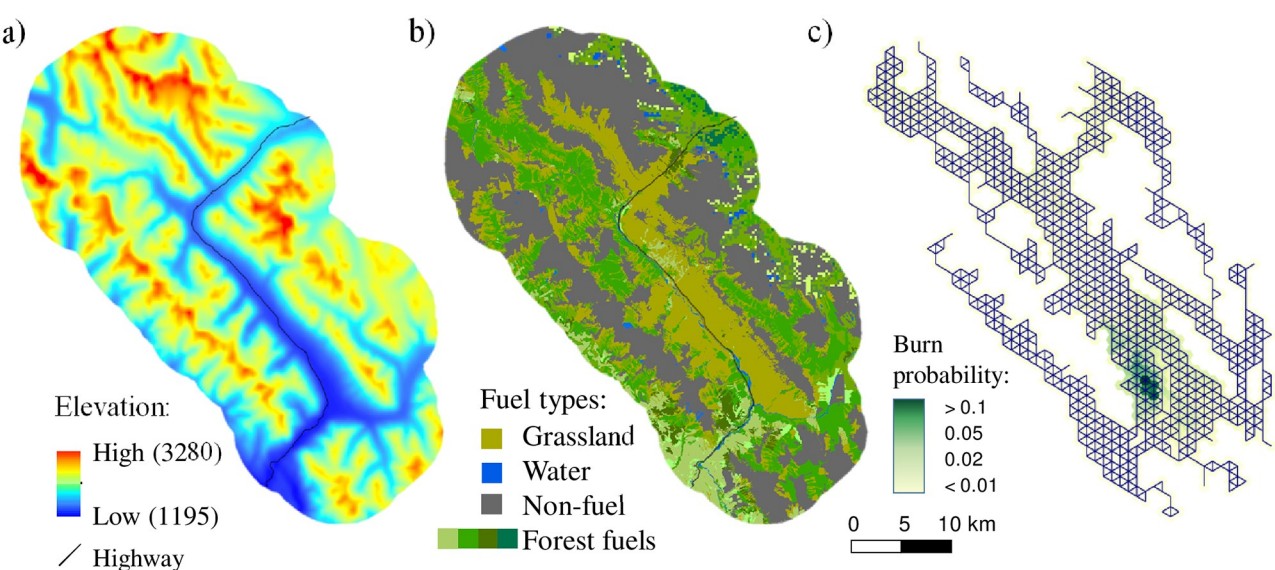

**Fig 4. a) Elevation; b) Fuel types; c) Landscape network G.** Dark blue lines depict edges *E* between nodes *N* in landscape network *G*. Dark green area denotes the hotspot with high burn probabilities based on Burn-P3 simulations.

surface fires are frequent but do not burn large areas [91]. Fire management in the park aims to balance the historical fire regime in fire-maintained ecosystems with the protection of the park's human infrastructure [93].

We applied the Burn-P3 model to simulate potential fire ignitions and spread in the Vermilion Valley area (approximately 834 km²) under the current management regime (see Supplement S1). We used the fire regime scenario from Reimer et al. [88] that assumed fire management crews attack all detected fires in the area but, on average, 13% of ignited fires would escape initial attack. Fire suppression has the greatest impact on fires that are small at the time of discovery and is substantially less effective when fires are already large. For example, Alberta reported 83% success at containing fires smaller than 3 ha [94] and British Columbia reported 92% success containing fires smaller than 4 ha [95].

A 5-km buffer was added to allow fires to spread in and out of the study area. We calibrated the model to align generally with the historical fire size distribution in the Vermilion Valley area from the Canadian National Fire Database [96] (Fig 5).

In Canada, complex terrain and poor access often necessitate the use of helicopters for wildfire management operations [41]. Compared to the use of ground vehicle-based engine crews, the use of helicopters increases the total cost for a burn treatment, but also keeps the average per-unit-area costs fairly consistent no matter where the prescribed burns are executed in the study area, thereby making implementation less dependent on the cost of ground access to a particular location. Hence, we assumed the total cost to be proportional to the prescribed burn area. Note that the budget constraint [2] would require a variable cost component if the treatments were managed by ground crews, in which case their cost would depend on the time needed to access the treatment sites (which, in turn, could be a function of the complexity of terrain and the proximity to roads).

Our landscape network *G* included 834 1-km² hexagonal nodes interconnected in a triangular pattern (Fig 4c). We chose this node size to balance the numeric complexity of the problem with its practical utility for planning. It was also an appropriate size for capturing the patterns of natural fire barriers in the area, such as individual mountain ridges or rocky

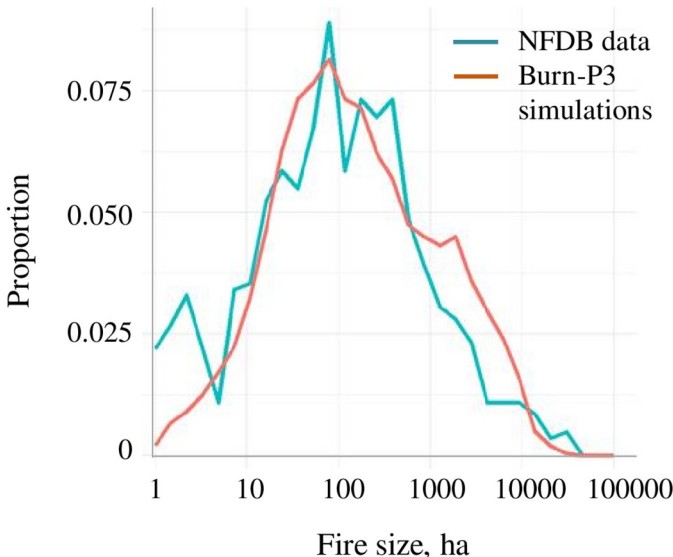

**Fig 5. Fire size distribution generated by Burn-P3 vs. the historical fire size distribution from the National Forest Fire Database.**

outcroppings that have been shown to be effective barriers to fire spread at a 250-m and greater scale in the park [97]. Based on this fixed node size, the treatment area limit $B$ defined the number of nodes to be removed from the network.

We found optimal solutions for problem 1 without spatial contiguity constraints as well as problems 2 (single-period) and 3 (multi-period) with spatial contiguity constraints for a range of treated areas between 10 and 50 nodes. The minimum and maximum sizes of prescribed burn $t$, $A_{min}$ and $A$, were set to 2 and 6 nodes in problems 2 and 3 and the treatment area limit $B$ was set to $5 \times T$ nodes. To determine what fire spread information adds to the optimal solution, we solved the original CND problem (1–4) using the current land cover composition and no $p_{ij}$ values (Fig 4c). This scenario was based solely on a binary map of flammable/non-flammable land cover types (e.g., vegetation and natural fire barriers). Alternatively, the scenarios that included fire behaviour information utilized the fire spread probabilities $p_{ij}$ calculated with the fire simulation model.

We also evaluated a scenario that preferentially treated the sites with the highest ignition probabilities. In this simple scenario, nodes were removed in descending order of fire ignition probability, subject to a treatment area constraint (2). For each node, we estimated the ignition probabilities from Burn-P3 fire model outputs.

Hard combinatorial complexity makes it difficult to find feasible solutions for problems 2 and 3 with the number of planning steps $T > 2$. For problem 2, we used the solutions with $T = 2$ to initialize a three-step problem ($T = 3$). In this three-step problem, we fixed the decision variables $x'_{it}$, and $u'_{ijt}$ in steps 1 and 2 to the solutions with $T = 2$ as a warm start. After saving the optimal solution with $T = 3$, we increased the $T$ value to four steps and solved the model again using $x'_{it}$, and $u'_{ijt}$ for $t = [1;3]$ from the solution with $T = 3$ as a warm start, and so on until we solved the model for a desired number of steps. We then used the set of decision variables from the last solution to warm start the full problem. Problem 3 was initialized from problem 2 solutions in similar fashion. The model was run for 72 hours or until reaching an optimality gap of 0.5%, whichever came first. The model was composed in the General Algebraic Modeling System [98] and solved with the GUROBI linear programming solver [99].

## Results

We determined the minimum number of Burn-P3 model iterations to stabilize the fire spread probabilities $p_{ij}$ and optimal solutions in the CND problem (Supplement S3). The node removal patterns in the CND model solutions stabilized after 30,000 iterations (S3 Fig 1 and 2 S3 File), so we report the optimal solutions for problems 1–3 using $p_{ij}$ values calculated from 30,000 Burn-P3 iterations (Table 2).

### Mapping the fire spread probabilities

The study area included a large concentrated area of high burn probabilities in the southern portion and a few smaller areas with elevated burn probabilities along the major highway crossing the study area (Fig 6a). The map of fire spread probabilities $p_{ij}$ between node pairs (Fig 6b) generally agreed with the burn probability map, but the large number of overlapping fire spread arcs $ij$ between node pairs has made it too complex for practical use. We devised an alternative mapping approach where we plotted the fire spread potential recalculated as the spread between adjacent nodes $i,j$, $w_{ij}$, along network edges $E$. We calculated the $w_{ij}$ values from the same set of Burn-P3 outputs we used to calculate the $p_{ij}$ values for the CND model (see Supplement S4). For each simulated fire, we calculated the fire spread vectors between adjacent nodes, starting from the fire ignition node, to all nodes within the fire perimeter via a shortest path algorithm (S4 Fig 1a in S4 File). Each vector $ij$ connecting adjacent nodes $i$ and $j$ was assigned one. For each arc, $ij$, $ij \in E$, we then summed the spread vectors calculated for all individual fires and divided this value by the number of Burn-P3 iterations. This gave us the fire spread probability between adjacent nodes $i$ and $j$. The $w_{ij}$ values were used only to map the fire spread patterns and were not utilized in the CND model.

The map of $w_{ij}$ values shows the fire spread probabilities along edges $E$ in landscape network $G$ (Fig 6c) and reveals some critical elements of fire spread in the area [100]: prevailing fire spread directions, when darker edges align in parallel streaks (Fig 6c callout I); areas with omnidirectional spread of fires, when a set of neighboring edges of similar color exhibits a triangular pattern (Fig 6c callout II); or potential fire spread corridors (Fig 6c callout III). We also plotted similar maps for node removal solutions (Fig 6d–6l), where we calculated the $w_{ij}$ values only for the fire spread paths between node pairs with non-zero products of decision variables $u_{ij}$ and fire spread probabilities $p_{ij}$, $p_{ij}u_{ij} > 0$ (S4 Fig 1b in S4 File). These maps help assess the impact of node removal on fire spread risk. Note that the $w_{ij}$ values were only used to visualize the fire spread patterns. The optimization model used the actual fire spread probabilities $p_{ij}$.

### Optimal network interdiction solutions

The panels of Fig 6d through Fig 6l show problem 1 solutions, which ignored spatial contiguity constraints for site treatments, for treatment area (i.e., node removal) limits $B$ = 10, 20 and 30 nodes. The solutions without fire spread information split the central Vermilion Valley almost in half (Fig 6d–6f), while the rest of the budget was spent on isolating small valleys connected to the central valley by narrow corridors. As the treatment area limit $B$ increased, a larger share of the budget was spent on isolating these small valleys (Table 2).

Incorporating information about fire spread changed the optimal node removal patterns. In the binary fireshed problem 1 solutions (Fig 6g–6i), the central Vermilion Valley was not cut in half as in the solutions without fire spread information. Instead, the immediate focus at lower budget levels was on isolating the smaller valleys. As the treatment area increased, the model fragmented the area with high risk of ignitions in the southern part of the Vermilion Valley (Fig 6i, Table 2).

Table 2. Node removal summaries in problem 1–3 optimal solutions.

| Problem/ scenario | Treatment area B, nodes | Planning steps, T in problem 2 (or periods, T' in problem 3) | Expected number of path connections between node pairs based on: | | Budget portion, %, spent on: | | |
|---|---|---|---|---|---|---|---|
| | | | $p_{ij\ bin}$* | $p_{ij}$** | 1-node segments | 2-3-node segments | >3-node segments |
| Problem 1—using land cover composition only | | | | | | | |
| | 10 | | 60494 | 22.4 | 30% | 20% | 50% |
| | 20 | | 49793 | 20.2 | 35% | 35% | 30% |
| | 30 | | 42578 | 19.3 | 33% | 47% | 20% |
| Minimizing the probability of ignitions | | | | | | | |
| | 10 | | 63440 | 20.7 | 10% | 90% | - |
| | 20 | | 58368 | 18.1 | 10% | - | 90% |
| | 30 | | 55696 | 16.4 | - | 10% | 90% |
| Problem 1 binary fireshed scenario | | | | | | | |
| | 10 | | 51982 | 20.6 | 50% | 50% | - |
| | 20 | | 41310 | 17.0 | 20% | 60% | 20% |
| | 30 | | 34400 | 15.4 | 20% | 67% | 13% |
| Problem 1 probabilistic fireshed scenario | | | | | | | |
| | 10 | | 55935 | 18.6 | 10% | - | 90% |
| | 20 | | 48077 | 15.4 | - | 30% | 70% |
| | 30 | | 43076 | 13.1 | - | 33% | 67% |
| Problem 2 binary fireshed scenario | | | | | | | |
| | 10 | 2 | 56485 | 20.0 | - | - | 100% |
| | 20 | 4 | 46624 | 17.4 | - | - | 100% |
| | 30 | 6 | 40049 | 15.1 | - | - | 100% |
| Problem 2 probabilistic fireshed scenario | | | | | | | |
| | 10 | 2 | 58452 | 18.8 | - | - | 100% |
| | 20 | 4 | 49444 | 15.7 | - | 10% | 90% |
| | 30 | 6 | 41731 | 13.2 | - | 7% | 93% |
| Problem 3 binary fireshed scenario | | | | | | | |
| | 10 | 2 | 56489 | 20.1 | - | - | 100% |
| | 20 | 4 | 46624 | 17.4 | - | - | 100% |
| | 30 | 6 | 40049 | 15.1 | - | - | 100% |
| Problem 3 probabilistic fireshed scenario | | | | | | | |
| | 10 | 2 | 58452 | 18.8 | - | - | 100% |
| | 20 | 4 | 49444 | 15.7 | - | 10% | 90% |
| | 30 | 6 | 41731 | 13.2 | - | 7% | 93% |

* Used in binary fireshed scenarios; Y-axis in Fig 10a; X-axis in Fig 11.

**Used in probabilistic fireshed scenarios; Y-axis in Fig 10b; Y-axis in Fig 11.

The probabilistic fireshed solutions differed from the binary fireshed solutions (Fig 6j–6l). Most of the budget was spent on partitioning the area with high risk of ignitions in the southern region. At larger treatment area levels, a small portion of the budget was allocated to creating one or two large partitions in the north. Compared to the binary fireshed scenarios, none

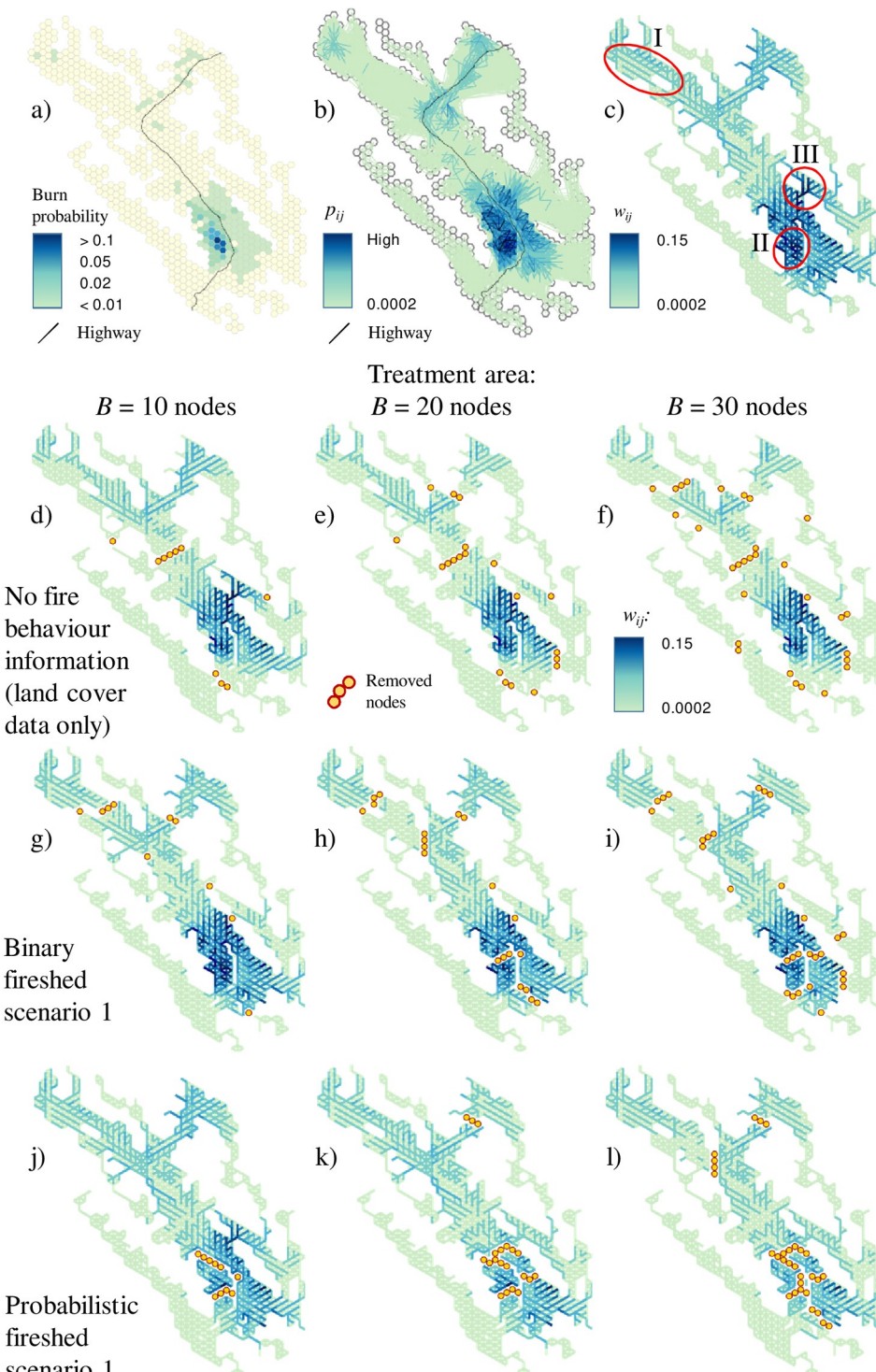

**Fig 6. Maps of optimal node removal solutions.** Key fire regime metrics: a) burn probabilities; b) fire spread probabilities between node pairs, $p_{ij}$ (spread probabilities along network edges $E$ between adjacent nodes are not shown); c) fire spread probabilities between adjacent nodes, $w_{ij}$. Node removal solutions based on land cover configuration only: d) treatment area $B = 10$ nodes; e) $B = 20$ nodes; f) $B = 30$ nodes. Binary fireshed scenario 1 solutions (using $p_{ij\ bin}$ values): g) $B = 10$ nodes: h) $B = 20$ nodes; i) $B = 30$ nodes. Probabilistic fireshed scenario (using $p_{ij}$ values): j) $B = 10$ nodes: k) $B = 20$ nodes; l) $B = 30$ nodes. Maps c-l use the same color scheme for plotting $w_{ij}$ values.

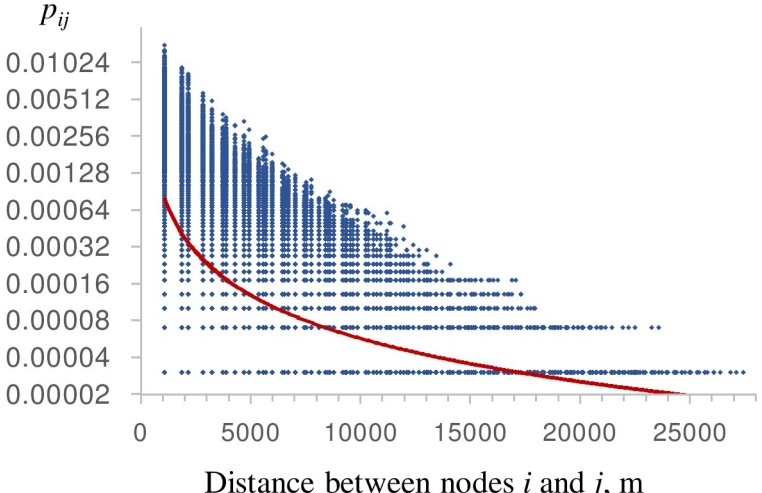

**Fig 7. Fire spread probability, $p_{ij}$, between a pair of nodes vs. distance between node centers.** Bold line is a power law approximation, $p_{ij} = 2.835 \cdot \text{distance}^{-1.174}$. Y-axis is shown in $\log_2$ scale.

of the budget was spent on disrupting the narrow corridors connecting smaller valleys. Overall, the probabilistic fireshed scenarios more effectively reduced the local spread of fires between adjacent nodes than the binary fireshed scenarios (Fig 6j–6l). This is unsurprising because the probabilistic scenarios minimize the spread of the most frequent, small fires, which tend to spread between adjacent nodes only. The relationship between fire spread distance and the probability of spread $p_{ij}$ is described by a power law function (Fig 7). When the unaltered $p_{ij}$ values were used in objective (5), hotspots with frequent small fires were prioritized for interdiction. By comparison, in the binary fireshed scenarios, all positive $p_{ij}$ values were converted to one, so there was no direct feedback from the spread probabilities to the objective value (as long as $p_{ij} > 0$). The number of node pairs with possible path connections grows in quadratic proportion to the linear distance between nodes; consequently, minimizing objective (5) in the binary fireshed scenarios minimizes long-distance fire spread.

We also compared the problem 1 solutions (Fig 6g–6l) with a scenario that minimized the risk of fire ignitions (Fig 8b–8d). Regardless of the treatment area limit, this strategy consistently removed nodes from the hotspot of high ignition probabilities in the southern part of Vermilion Valley but did not effectively fragment the landscape (Table 2).

Controlling the spatial contiguity (problems 2 and 3) and timing (problem 3) of prescribed burns moderately changed the node removal patterns but the spatial allocation generally followed the patterns in the problem 1 solutions (Fig 9). Because the node removal patterns in the problem 2 and 3 solutions were similar, the patterns of removed nodes are shown for problem 3 solutions only. A key difference between problems 2 and 3 is that problem 3 tracks the cumulative impact of node removal over time by solving the CND sub-problem for each period $t'$, whereas problem 2 solves only one CND sub-problem for the entire planning horizon. Adding a temporal dimension aligns node removal with a first-best strategy when the prescribed burn that causes greatest reduction in fire spread potential is scheduled first, then the second most impactful burn is planned and so on. The impact of nodes removed in period 1 lasts for $T'$ periods, $T'-1$ periods for nodes removed in period 2 and so on. The solutions for problems 2 and 3 diverge slightly in large-budget scenarios as the number of planning steps

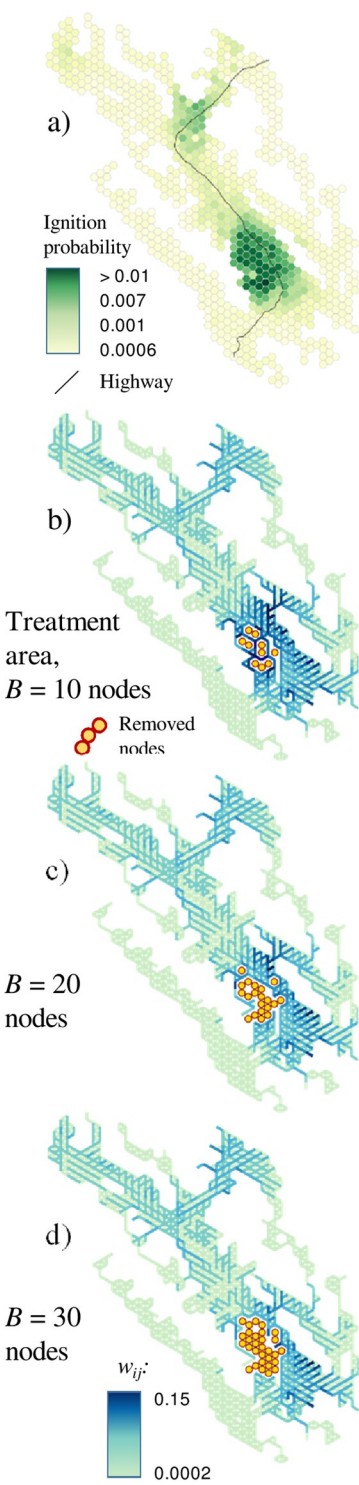

**Fig 8. Optimal solutions minimizing the probability of ignitions: a) node-based ignition probabilities; b) solutions at treatment area level *B* = 10 nodes: c) solutions at *B* = 20 nodes; d) solutions at *B* = 30 nodes.**

increases (Fig 10). Problem 3 is more realistic because it provides guidance to forest managers about a stepwise implementation of the fuel reduction strategy.

In binary fireshed solutions, nodes were removed to isolate parts of the study area north of the highway and, as the treatment area increased, partition the hotspot of high burn probabilities in the southern portion of the area (Fig 9a–9c). In probabilistic fireshed solutions, the bulk of the budget was spent on fragmenting this hotspot (Fig 9d–9f). Given that the probabilistic fireshed solutions for problem 1 also prescribed the removal of large node segments in the hotspot area (Fig 6j–6l), adding the contiguity constraints in problems 2 and 3 caused relatively minor changes in the node removal patterns.

Fig 10 depicts the objective value as a function of the total treatment area. All optimal solutions had zero penalties $V_t$, so the objective values for both problem 1 and 2 solutions are shown as $\sum\limits_{i=1}^{i\in N}\sum\limits_{j\neq i}^{j\in\Omega_i} u_{ij}p_{ij}$ and for problem 3 solutions as $\sum\limits_{i=1}^{i\in N}\sum\limits_{j\neq i}^{j\in\Omega_i} u'_{ijT'}p_{ij}$, which depicts the node connections at the end of horizon $T'$. Note that the actual objective values for problem 3 are worse than they appear in Fig 10 because the objective function (17) track the gradual reduction of fire spread potential over timespan $T'$.

The probabilistic and binary fireshed scenarios performed as expected for their specific objectives. However, they performed poorly with respect to the alternate objectives. For

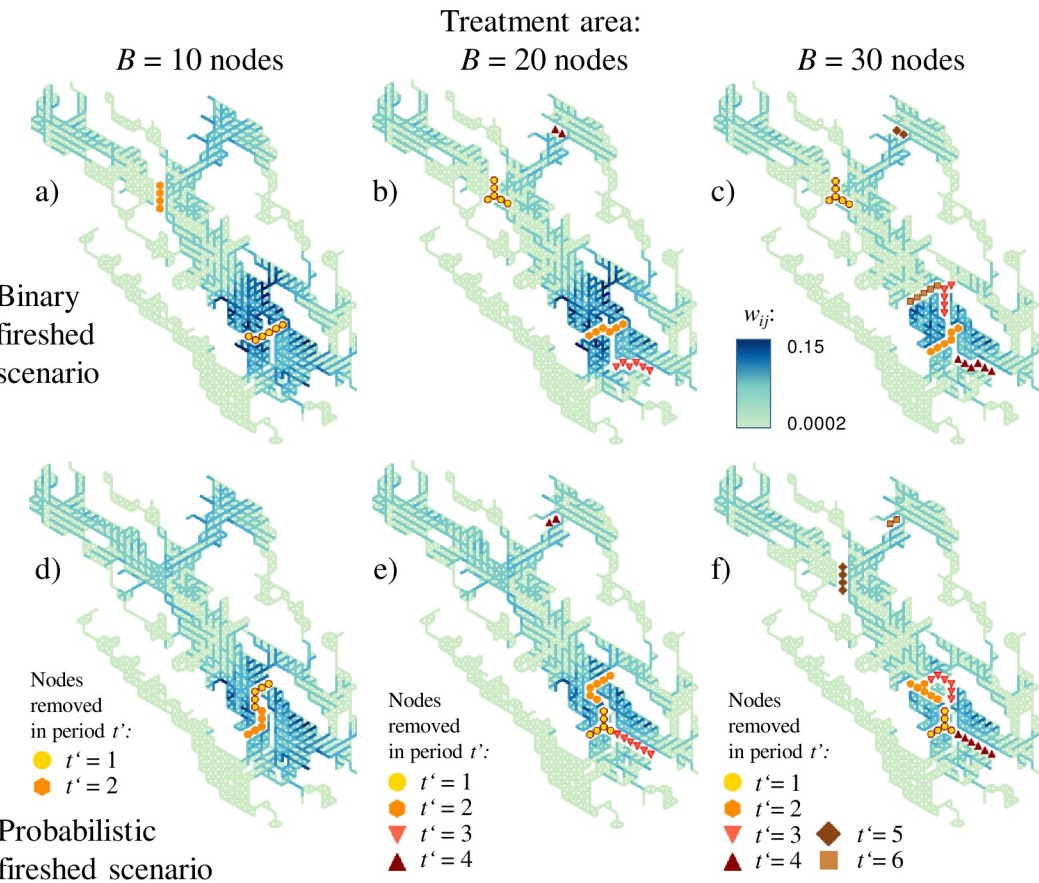

**Fig 9. Optimal problem 3 solutions.** Binary fireshed scenario using $p_{ij\ \text{bin}}$ values: a) treatment area $B = 10$ nodes, two periods; b) $B = 20$ nodes, four periods; c) $B = 30$ nodes, six periods. Probabilistic fireshed scenario using $p_{ij}$ values: d) $B = 10$ nodes, two periods; e) $B = 20$ nodes, four periods; f) $B = 30$ nodes, six periods.

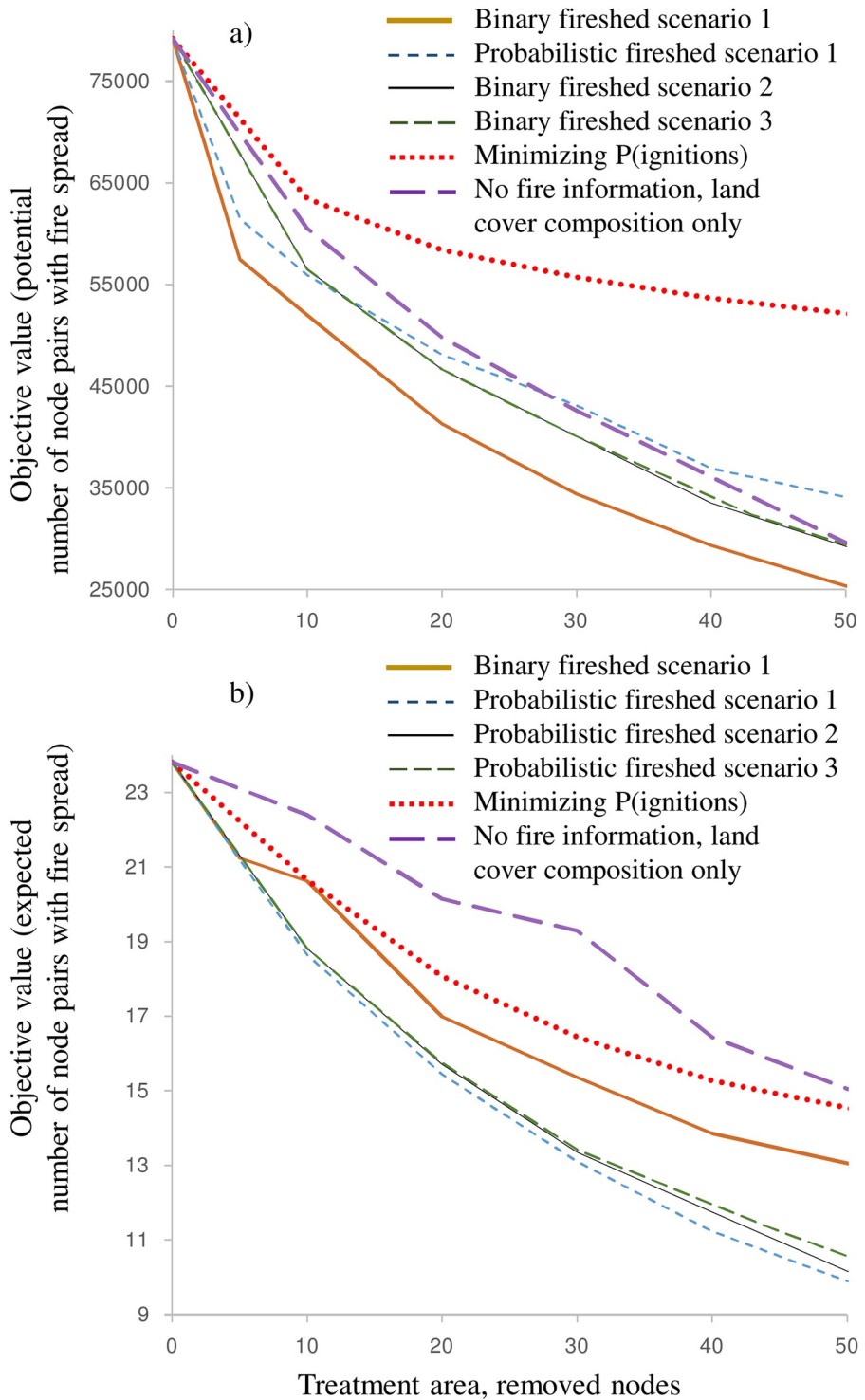

**Fig 10. Objective value vs. total treatment area, nodes: a) binary fireshed objective with $p_{ij\text{ bin}}$ values; b) probabilistic fireshed objective with $p_{ij}$ values.** Lower objective values show better outcomes. Objective values for problem 3 solution are shown as $\sum_{i=1}^{i\in N}\sum_{j\neq i}^{j\in\Omega_i} u'_{ijT'}p_{ij}$ which denotes the fire spread potential at the end of timespan $T'$.

example, the probabilistic fireshed scenario performed poorly in terms of the binary fireshed scenario objective, $\sum\limits_{i=1}^{i \in N} \sum\limits_{j \neq i}^{j \in \Omega_i} u_{ij} p_{ij\text{min}}$ (Fig 10a) and vice versa, the binary fireshed scenario demonstrated poor performance in term of the probabilistic fireshed scenario objective, $\sum\limits_{i=1}^{i \in N} \sum\limits_{j \neq i}^{j \in \Omega_i} u'_{ij} p_{ij}$ (Fig 10b). The performance gap increased as the total treatment area increased.

Thus, the choice of the objective (i.e., either minimize the spread of large fires or frequent small fires) greatly impacts the optimal solutions.

The curves in Fig 10 generally follow the rule of diminishing returns: the largest marginal reduction of fire spread potential is achieved at small budget levels. In binary fireshed solutions, the impact of adding contiguity constraints (and temporal dimension in problem 3) was noticeable across the whole budget range (Fig 10a). Notably, the scenario that minimized the probability of ignitions was ineffective at reducing the spread of large fires (Fig 10a, Table 2).

The impact of adding the spatial contiguity constraints and temporal dimension was less noticeable in the probabilistic fireshed solutions (Fig 10b). This is because all scenarios tended to remove nodes in large segments from the hotspot region with the highest burn probabilities (Figs 6j–6l and 9d–9f). In short, the probabilistic fireshed scenarios focused on the area with the most frequent fires (which, as noted, are typically the smallest). The solutions minimizing the probability of ignitions were not as effective as the problems 1–3 fireshed solutions (Fig 10).

## Preventing the spread of frequent small fires vs. large rare fires

We depicted the trade-off between the binary and probabilistic fireshed scenarios in dimensions of their objective values (Fig 11). Single-period problem 1 solutions demonstrated the best performance with respect to both objectives (Fig 11 callout 1). Adding the burn contiguity constraints and temporal dimension in problems 2 and 3 worsened the objective values (Fig 11 callouts II, III). The spatial contiguity requirement worsens the objective value because it prevents the allocation of single-node treatments. These burns are most cost-effective in small-budget solutions, but multiple small burns are difficult to implement in practice. Problem 1–3 solutions performed better across than the solutions minimizing the probability of ignitions or based on land cover composition only (Figs 10 and 11).

Combining the node removal strategies of the binary and probabilistic fireshed scenarios would achieve a balance between minimizing the spread of small and large fires (Fig 11, Supplement S5 Fig 1 in S5 File). The end points of the trade-off curves in Fig 11 depict the strategies preventing the spread of either the largest fires (binary fireshed solutions) or the smallest, most frequent fires (probabilistic fireshed solutions). However, the maps in S5 Fig 1 in S5 File depict examples of solutions on the trade-off curves that fall between these end points. For practical selection of suitable trade-off solutions, decision-makers would need to set the priority for a particular range of fire sizes that they would like to target. To accomplish this, the fire spread probabilities $p_{ij}$ for each pair of locations $i$ and $j$ potentially could be adjusted in the objective equation by a user-defined scaling coefficient based on the distance between $i$ and $j$. This would make the approach adaptable to other fire regime conditions and management objectives. The scaling coefficients could also be location specific. For example, protection against frequent small fires could be prioritized close to human settlements while simultaneously minimizing the spread of large fires in remote areas.

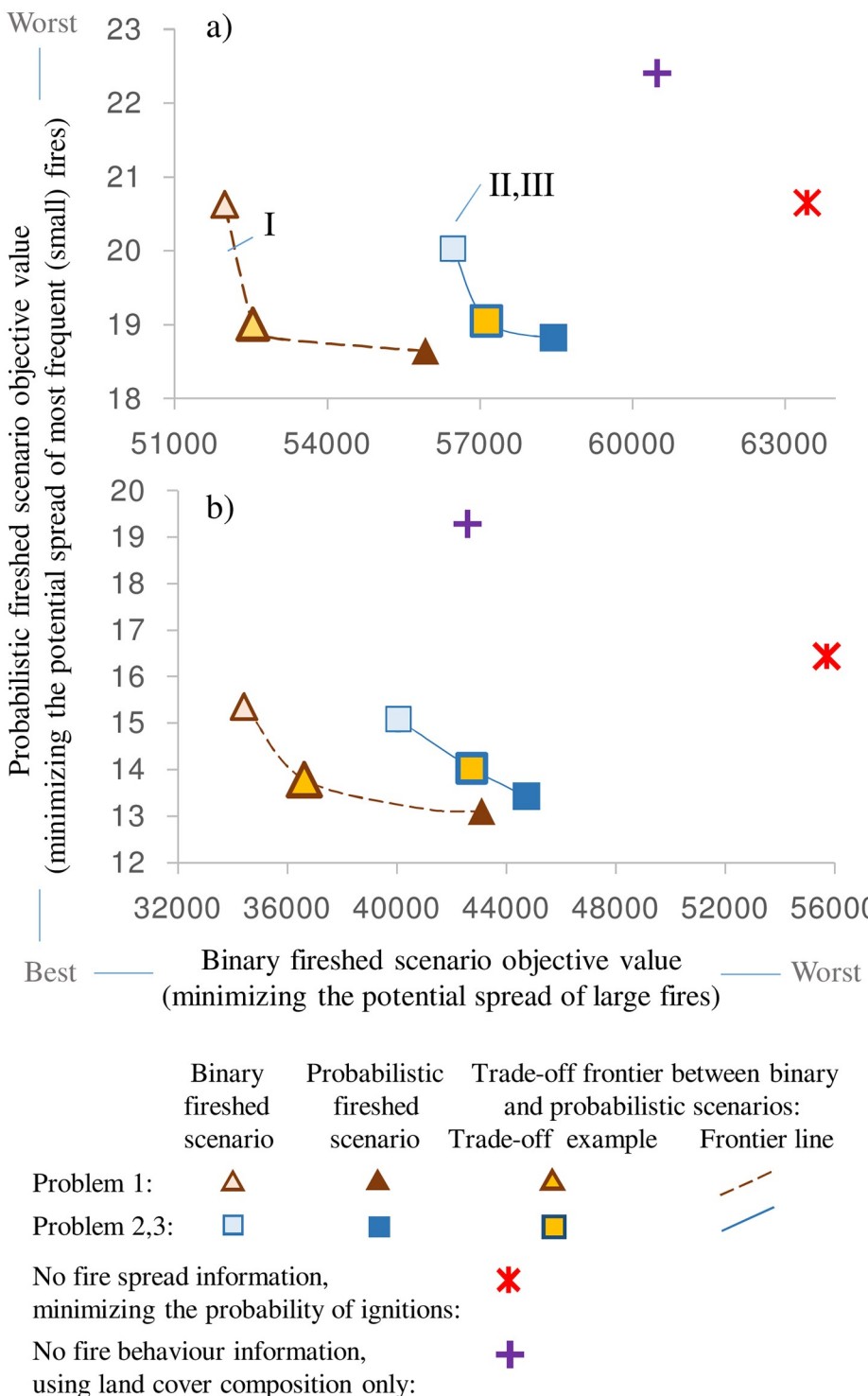

Fig 11. **Trade-off between problem 1–3 solutions in dimensions of binary and probabilistic fireshed scenario objectives: a) treatment area, *B* = 10 nodes; b) *B* = 30.** Trade-off frontiers between binary and probabilistic fireshed scenario objectives: I—problem 1 solutions; II, III—problem 2 and 3 solutions (appear close, so only problem 3 solutions are shown). Objective values for problem 3 solution are shown as $\sum_{i=1}^{i \in N} \sum_{j \neq i}^{j \in \Omega_i} u'_{ijT'} p_{ij}$ which denotes the fire spread potential at the end of timespan *T'*.

## Discussion

The proposed approach builds on previous work on optimizing wildfire fuel treatments in the following ways. First, we have addressed the chief limitation of using a node-based fuel load (or other fire hazard) metric, which is its inability to account for factors controlling directional spread of fires, such as prevailing winds and local weather, that may promote the spread of fires over long distances. The use of directional fire spread probabilities $p_{ij}$ helps us deal with this shortcoming. Our approach is an improved adaptation of the classical CND problem for lattice-type networks (which is a common way of depicting forest landscapes) because it enables application of the CND model in landscapes with few or no natural fire barriers. In this situation, node removal is guided by differences in the fireshed spatial configurations around each node (i.e., subsets $\Omega_i$) and the corresponding differences in the fire spread probabilities $p_{ij}$. Finally, our approach utilizes the full potential of stochastic fire simulation models by calculating the fire spread probabilities directly from the simulated perimeters and ignition locations of individual fires. The use of sophisticated spatial fire simulators (like Burn-P3) helps account for a multitude of factors influencing the spread of fires in highly heterogeneous landscapes. In addition to minimizing the probability of fire spread, the objective function could incorporate other fire behaviour parameters (such as fire intensity) in conjunction with spread.

Our approach is distinct from previously proposed fuel treatment strategies. One common strategy is to treat sites with the highest ignition probabilities (as in the simple scenario that we compared to our solutions in Fig 8). However, this strategy does not optimally reduce the risk of spread of escaped fires, nor does it address the uncertainty of determining which sites have the highest fire ignition potential. By comparison, our probabilistic fireshed strategy compartmentalizes regions with high ignition potential, thereby providing a hedge against the possibility of fires escaping to spread elsewhere.

Several other fuel treatment strategies have used site-specific priority weights. Minas et al. [23] linked site treatment and deployment resources to minimize the number of sites covered by these activities. Each site was assigned a weight by ignition probability and the value under risk if a fire originating in that site is not contained by the initial response. Rachmawati et al. [101] focused on rapid fuel accumulation after treatment and used site-based combinations of vegetation type and age since fire to find an optimal multi-period sequence of fuel treatments. Wei [21] applied optimization of fuel treatment at a very small scale (7×7 rows) without embedding a fire simulation model but examined the geometry of the treated areas. Finney et al. [102,103] proposed the assessment of fuel treatments by dividing the landscape into rectangular strips oriented normal to the predominant wind directions. Then, fire growth was simulated, starting with the strip farthest upwind, to identify key fire spread routes and their intersections with potential treatment areas. The process was repeated after moving each strip in the direction of the wind to impact downwind travel routes and subsequent treatment areas. This method finds fuel treatment configurations for a set of likely fire spread routes but overlooks the combinatorial aspects when allocating multiple treatments under a limited budget. Another network-based approach aimed to minimize the connectivity between sites with high fuel loads [24]. Pais et al. [14] used a network flow model to control the spatial contiguity of the treated area and prioritized treatments using a site-based fire risk metric (the Downstream Protection Value, DPV). The DPV metric assigns treatment priority ranks to sites by modeling fire propagation through a forest landscape as a tree graph and accounts for the potential of a fire ignited at a given site to burn other sites. In contrast, our model makes decisions using the fire spread probabilities between pairs of locations, which enables control based on the presence of possible fire spread paths between these locations. This makes the CND formulation

more effective in fragmenting the landscape and thus reducing the risk of substantial fire expansion than approaches that use site-based fire hazard metrics, but the problem is more complex numerically.

The CND model can be applied for multi-period planning in two ways. Single-period problems 1 and 2 could be solved in sequence. For each period, the manager plans a small number of prescribed burns and before the next period the fuel map is adjusted to account for the effect of these burns (as well as any changes in forest composition). This strategy does not consider the optimal timing of site treatment actions, which is addressed in a multi-period problem 3 formulation. However, the model for problem 3 has higher combinatorial complexity because it solves the CND problem for each planning period and tracks the cumulative impact of node removals over time, so the model may only be tractable for short planning horizons. Alternatively, problem 3 can be initialized from problem 2 optimal solutions and solved with constraint (21) to find an optimal timing of burns prescribed by the problem 2 solutions. While this approach does not guarantee the multi-period optimum, the solutions are likely to be close to the problem 3 optimal solutions, especially when the number of planned burns is small. For a relatively short timespan (like in our study), a time sequence of $p_{ij}$ values can be estimated with a fire simulation model prior to optimization, but this approach will not account for uncertain changes in fire occurrence that could emerge during the timespan. For longer planning periods and/or large landscapes, solving a simpler problem 1 or 2 in sequence with the recalculation of the fuel map and $p_{ij}$ values after each planning step may be more practical.

## Future model extensions

The CND problem is known to be NP-hard on general graphs [36,39,43]. One way to reduce the problem size is to limit the extent of the area where site treatments at an anticipated budget level could be cost-effective. Potentially, broad-scale regions where node removal could be cost-effective can be delineated by solving problem 1 for a range of treatment areas that slightly exceed the anticipated treated area. In turn, these regions can be applied as masks to restrict the extent of node removal selections in problems 2 and 3.

The proposed problem formulation minimizes the probability of fire spread across the landscape without specifying a particular direction of spread or locations of concern. As hinted at earlier, the problem could be reformulated to examine strategies for protecting human infrastructure from wildfires. In this case, one would only need to consider the fire spread paths (along with the corresponding pairs of nodes $u_{ij}$ and spread probabilities $p_{ij}$) that could potentially reach the area of concern. Instead of tracking the fire spread probabilities $p_{ij}$, the objective function could track the probabilities of a fire arriving from an ignition point $i$ to point $j$ within a given duration of time. This may require introducing decision variables that track, for each pair of locations, whether a fire ignited in location $i$ could arrive at location $j$ within a specified timespan, e.g., in a similar way to the formulation in Wei [21]. This could be a useful modification for planning treatments to protect human infrastructure, where tracking the fire arrival times is critical for evacuation planning.

Our study used a mountainous forest landscape to demonstrate the CND model. The approach could be applied in other landscapes (e.g., semiarid areas or areas where forest and agriculture are intermixed), as well as to fuel management strategies different than prescribed burns [104–106], as long as simulation models capable of producing realistic fire ignition and spread patterns are available. The problem could also be modified to evaluate alternative fuel treatment methods (e.g., prescribed burns vs. strategic thinning of forest stands) or different

management regimes (e.g., treatments of vehicle-accessible sites by ground crews vs. treatments from helicopters).

In our study, we used the ignition points and perimeters of the simulated fires to calculate the probabilities of fire spread between pairs of locations but did not track specific spread paths within individual fires. Tracking daily or hourly fire spread within a fire perimeter could help refine the spread probability values for long spread distances. Applying such model enhancements could be a focus of future work.

## Supporting information

**S1 File. Burn-P3 model inputs.**
(PDF)

**S2 File. Calculating the fire spread probabilities $p_{ij}$.**
(PDF)

**S3 File. Optimal solutions of the critical node detection problem vs. the number of Burn-P3 iterations.**
(PDF)

**S4 File. Calculating the fire spread probabilities $w_{ij}$ between adjacent nodes in a landscape network.**
(PDF)

**S5 File. Optimal solutions combining the node removal strategies of the binary and probabilistic fireshed scenarios.**
(PDF)

## Acknowledgments

Jane Park and David Tavernini from Parks Canada contributed helpful discussions on the methodology and early draft.

## Author Contributions

**Conceptualization:** Denys Yemshanov, Daniel K. Thompson, Marc-André Parisien.

**Data curation:** Ning Liu, Quinn E. Barber, Jonathan Reimer.

**Formal analysis:** Denys Yemshanov, Ning Liu, Daniel K. Thompson, Marc-André Parisien, Quinn E. Barber, Frank H. Koch.

**Funding acquisition:** Denys Yemshanov.

**Investigation:** Denys Yemshanov, Marc-André Parisien, Quinn E. Barber.

**Methodology:** Denys Yemshanov, Daniel K. Thompson, Frank H. Koch.

**Validation:** Denys Yemshanov, Daniel K. Thompson, Marc-André Parisien, Quinn E. Barber.

**Visualization:** Ning Liu.

**Writing – original draft:** Denys Yemshanov.

**Writing – review & editing:** Denys Yemshanov, Daniel K. Thompson, Marc-André Parisien, Quinn E. Barber, Frank H. Koch, Jonathan Reimer.

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
