## [Decision Letter · Decision Letter 0]

17 Jun 2021

PONE-D-21-15640

DETECTING CRITICAL NODES IN FOREST LANDSCAPE NETWORKS TO REDUCE WILDFIRE SPREAD

PLOS ONE

Dear Dr. Yemshanov,

Thank you for submitting your manuscript to PLOS ONE. After careful consideration, we feel that it has merit but does not fully meet PLOS ONE’s publication criteria as it currently stands. Therefore, we invite you to submit a revised version of the manuscript that addresses the points raised during the review process.

Both reviewers have indicated that the manuscript requires revisions to the Discussion, specifically that you should compare your method against existing methods to show the improvements or disadvantages. Reviewer 2 has indicated that the length of the methods should be reduced, but expects additions around describing the study area and Burn-P3 parameterization. Reviewer 1 has offered some specific suggestions for improvements in the methods section and the authors should consider moving some text to supplementary materials as suggested by Reviewer 2. Though I found the methods to be a lot to wade through, much of it is necessary in my mind. When deciding what to add or remove from the methods, the authors should be mindful to ensure that the manuscript meets PLOS ONE's criteria for publication, namely criterion 3:

3. Experiments, statistics, and other analyses are performed to a high technical standard and are described in sufficient detail.

We look forward to receiving your revised manuscript.

Kind regards,

Paul Pickell, Ph.D.

Academic Editor

PLOS ONE

Journal Requirements:

2. We note that Figure 3 in your submission contain map images which may be copyrighted. All PLOS content is published under the Creative Commons Attribution License (CC BY 4.0), which means that the manuscript, images, and Supporting Information files will be freely available online, and any third party is permitted to access, download, copy, distribute, and use these materials in any way, even commercially, with proper attribution. For these reasons, we cannot publish previously copyrighted maps or satellite images created using proprietary data, such as Google software (Google Maps, Street View, and Earth). For more information, see our copyright guidelines: http://journals.plos.org/plosone/s/licenses-and-copyright.

2.1.    You may seek permission from the original copyright holder of Figure 3 to publish the content specifically under the CC BY 4.0 license. 

2.2.    If you are unable to obtain permission from the original copyright holder to publish these figures under the CC BY 4.0 license or if the copyright holder’s requirements are incompatible with the CC BY 4.0 license, please either i) remove the figure or ii) supply a replacement figure that complies with the CC BY 4.0 license. Please check copyright information on all replacement figures and update the figure caption with source information. If applicable, please specify in the figure caption text when a figure is similar but not identical to the original image and is therefore for illustrative purposes only.

Reviewers' comments:

Reviewer's Responses to Questions

**Comments to the Author**

1. Is the manuscript technically sound, and do the data support the conclusions?

Reviewer #1: Yes

Reviewer #2: Yes

2. Has the statistical analysis been performed appropriately and rigorously? 

Reviewer #1: Yes

Reviewer #2: Yes

3. Have the authors made all data underlying the findings in their manuscript fully available?

Reviewer #1: Yes

Reviewer #2: Yes

4. Is the manuscript presented in an intelligible fashion and written in standard English?

Reviewer #1: Yes

Reviewer #2: Yes

5. Review Comments to the Author

Reviewer #1: General comments:

Your work is interesting and, for the most part, well-presented. I find no major flaws in your analysis that warrant revision to the methods, but there are a few sections of the paper that could be improved to clarify your assumptions and methods. I’ll highlight my main critiques here, followed by specific recommendations.

First, I suggest you revisit the abstract to make sure it aligns with what you accomplished. Specific notes follow.

Fire behavior is a general term that includes spread but also intensity, duration, and type. In most cases, I think it would be clearer to say something like burn probability or spread likelihood instead in this paper.

Does the problem 3 formulation with time add value to your analysis? As you point out, you do not actually remodel the fire spread probabilities. My personal view is that there is little value planning far in advance in fire-prone landscapes because stochastic wildfire activity will alter the priorities for subsequent periods more than the treatments. If you are interested, other researchers have examined the effects of uncertainty in fire occurrence using two stage models solved with backwards induction:

Konoshima M, Montgomery CA, Albers HJ, Arthur JL (2008) Spatial-endogenous fire risk and efficient fuel management and timber harvest. Land Economics 84(3), 449-468.

Konoshima M, Albers HJ, Montgomery CA, Arthur JL (2010) Optimal spatial patterns of fuel management and timber harvest with fire risk. Canadian Journal of Forest Research 40, 95-108.

These should probably be cited in the discussion when discussing multiple planning periods. I’m surprised you advocate for problem 3 formulation in the discussion given that you say it produced similar results as problem 2 but with added complexity. Why?

If you had more space, I would recommend you compare your method to something simple and widely used, such as Finney’s treatment optimization method or specific patterns of fuel treatments. When you start to describe the contexts that your problem 3 can be applied in (small landscapes, short durations), I start to wonder if you really diverge from a plan that interrupts the major spread paths of an anticipated problem fire scenario.

Finney MA (2004) Chapter 9, Landscape fi re simulation and fuel treatment optimization. In: J.L. Hayes, A.A. Ager, J.R. Barbour, (tech. eds). Methods for integrated modeling of landscape change: Interior Northwest Landscape Analysis System. PNW-GTR-610. p 117-131.

Finney MA (2001) Design of regular landscape fuel treatment patterns for modifying fire growth and behavior. Forest Science 47(2), 219-228.

Specific comments:

Lines 40-42: It is fair to point out that resources to mitigate wildfire risk are limited, and it is therefore important to prioritize, but there are many tools to assist forest managers in planning fuel treatments. I suggest dropping the focus on limited tools here. Whether the tools are used by managers is another issue.

Line 42: I suggest replacing “interdiction” with common language since you felt the need to define it in the paper.

Line 43: Did you really cover consequences in this work? I didn’t see an effects analysis here.

Lines 44-45: How about “We used simulation modeling to estimate the likelihood of fire spread between forest network nodes and we…”

Line 46: fuel treatment ALLOCATION problem

Lines 64-68: Yes. Costs tend to rise with fire sizes, but rising suppression costs are often attributed to the expansion of human values into wildlands. It would be less controversial if you focused this statement only on the challenge of suppressing fires in rugged landscapes.

Lines 69-70: I think it is appropriate to at least mention that not all forestry and fire scientists agree that fuel treatments will reduce fire spread. Your Agee reference on shaded fuelbreaks includes discussion of this. Agee and Skinner (2005) and Reinhardt et al. (2008) argue strongly that most fuel treatments are, or should be, aimed at reducing fire intensity and severity.

Lines 70-72: Is the saving costs statement supported by these citations? It was a modeling study, but the Thompson et al. (2013) ‘Quantifying the Potential Impacts of Fuel Treatments on Wildfire Suppression Costs’ article provides the clearest estimates of how fuel treatments could reduce costs via their effects on fire sizes.

Lines 76-77: Again, I think it is appropriate to acknowledge that some of these models were aimed at reducing the severity of effects instead of large fire spread.

Lines 115-116: I would drop the subscripts here and save them for the methods.

Lines 127-130: Again, I think you should temper this statement to make it clear that it is more of an assumption supported by rules of thumb than a clear conclusion of the research. It is also prudent to acknowledge that fuel treatments do not generally achieve 100% protection, especially in the case of extreme weather. The Kalies and Kent (2016) review on fuel treatment effectiveness may be worth mentioning here.

Line 134: You already introduced the CND abbreviation.

Line 156-157: Would it not be simpler to introduce the model as area limited since you don’t account for variable costs? I see the future value of accounting for this, but it adds slight confusion to the paper. For example, you describe the model as having an upper bound for Rx fire area in the abstract and introduction.

Lines 183-185: And fire suppression?

Line 197: “depict well” to “represent”?

Line 234: “Consecutively” or “consequently”?

Lines 313-314: I do not think it is a good idea to use T and t for both steps in problem 2 and time periods in problem 3. I suggest changing a different letter for the steps in problem 2 to avoid confusion.

Lines 405-407: Rephrase for clarity.

Lines 437-442: You should clarify exactly how the information you mention was used. Were treatments limited to a particular vegetation type? Did you use the ignition probability, but not spread probability components of Burn P3? Is this later what you refer to as prioritizing on ignitions?

Lines 443-452: This is where it would help to know the difference between T steps and T time periods.

Lines 464-474: As noted in my general comments, this is an important enough change in methods that you should clarify which results it applies to (all?) and describe it fully in the methods section instead of supplementary material referenced from the results.

Lines 488-489: “ignoring spatial contiguity rules”? or “ignoring the simulated connectivity measures”?

Line 494: Suggest changing fire behavior to fire spread.

Lines 520-525: Why is this scenario suddenly popping up in the results? This should be introduced earlier with justification for what it tells you. Prescribed fires likely reduce ignition risk for a short period after treatment, but this will not last long as fuels reaccumulate. Reducing ignitions with rules and enforcement may require different methods in some landscapes.

Lines 579-582: I’m confused about what scenarios are being compared here.

Lines 598-611: I’m wondering how much the small/large fire size tradeoff that is important at this site pertains to the use of probability vs. binary fireshed weighting versus the specific pattern of fire sizes and occurrence on your landscape? What do you think you would find on a landscape with high probability of fire spread from less frequent but large fires?

Lines 615-619: Did the approximation you made to get at spread paths within fires really “address” the problem? I don’t have a brilliant solution to do better without complicating the simulation. I would be tempering my language here to reflect that some approximations were made to prototype a model framework.

Line 625: With a shortest path approximation…

Figure 3: You should probably include a scale bar and north arrow in the study site panel.

Reviewer #2: GENERAL COMMENT

The manuscript entitled “Detecting critical nodes in forest landscape networks to reduce wildfire spread” aims to propose a modeling approach to optimize preventive fuel treatments for minimizing the wildfire spread likelihood and consequences. The methodological approach presented in this manuscript was tested in a study area located in Kootenay National Park, British Columbia (Canada).

Overall, I do think the work is interesting and has the potential to provide insights and methods for future studies or analysis that would investigate the potential effects of spatial locations of fuel treatments on wildfire spread, while considering the maximization of the benefit/cost ratio. The present study could also provide relevant information for policy makers and stakeholders to adapt or improve future management plans and strategies in Canada as well as in other areas.

The Introduction section is well written and provides a generally good overview of the works that investigated this topic. I only have a remark. The authors limit the Introduction section focusing on previous works carried out in forest areas and using prescribed burnings, while the applicability of the approach they propose could be expanded also to semiarid or rural areas, as well as to fuel management strategies different than prescribed fires (see for instance, among others, Archibald et al. 2005, https://esajournals.onlinelibrary.wiley.com/doi/abs/10.1890/03-5210; Davies et al. 2015, http://dx.doi.org/10.1071/WF15055; Salis et al. 2018, https://www.sciencedirect.com/science/article/pii/S0301479718301191; Prichard et al. 2020 https://pubmed.ncbi.nlm.nih.gov/32086976/ doi:10.3390/f6062148). This is a shortcoming that can be improved.

The section related to Material and Methods is on the whole complete but suffers from the excessive length of the text (from L122 to L384, to L452 including the Case Study Description). Even if manuscripts published in Plos One can be any length, there is need to reduce this part and omit some redundant sentences. Some specific points to improve this section will be provided in later rows.

The Results are in my opinion fine. I would suggest making some improvements in Figures 3 and 4.

The Discussion section needs to be improved, as the comparison between the results and approach presented in this work are not compared with those obtained in other similar works.

In the end, the manuscript is overall of good quality and well-written, but should be improved by reducing the length of the text and improving the quality of the discussion sections.

SPECIFIC COMMENTS

Introduction: The Introduction section seem too much focused on works carried out in forest areas and application of prescribed burnings, while the approach proposed in this study could be expanded also to semiarid or rural areas, as well as to fuel management strategies different than prescribed fires (see for instance, among others, Archibald et al. 2005, https://esajournals.onlinelibrary.wiley.com/doi/abs/10.1890/03-5210; Davies et al. 2015, http://dx.doi.org/10.1071/WF15055; Salis et al. 2018, https://www.sciencedirect.com/science/article/pii/S0301479718301191; Prichard et al. 2020 https://pubmed.ncbi.nlm.nih.gov/32086976/ doi:10.3390/f6062148).

L386-389: Please include the size of the study area, as well as the total size of the modeling domain (that is, including the buffer area)

L399-402: Considering that no information is provided on crown fire and spot fire settings, I suppose the authors applied Burn-P3 model to simulate surface fire spread. In case the authors simulated crown fires and spot fires, I would recommend including more details on this.

L408-410: Please include a table, in the Supplementary data, to summarize the main input data used for fire simulations (e.g.: weather scenarios tested, wind directions and speed, fire spread durations, etc.)

L435-439: Problems 1-4 related to the critical nodes detection (CND) were introduced in the first equations, several pages before this part. I would recommend helping readers and clarifying that these problems refer to the first equations and the detection of critical nodes.

L533: Starting a sentence with “Recall that” might be inappropriate, please check

L613-666: The Discussion section summarizes relatively well the principles and generalizations from results as well as the significance of results. On the other hand, it does not discuss the results and methods presented in this work in relation to those of others. This is a limitation, so I recommend improving the Discussion in this sense.

Figures 3-4: Please include the scale bar.

6. PLOS authors have the option to publish the peer review history of their article (what does this mean?). If published, this will include your full peer review and any attached files.

Reviewer #1: No

Reviewer #2: No

---

## [Author Response · Author response to Decision Letter 0]

5 Aug 2021

PONE-D-21-15640 “DETECTING CRITICAL NODES IN FOREST LANDSCAPE NETWORKS TO REDUCE WILDFIRE SPREAD” – A Reply to Reviewers’ Comments.

Academic Editor’s comments:

Both reviewers have indicated that the manuscript requires revisions to the Discussion, specifically that you should compare your method against existing methods to show the improvements or disadvantages.

We have added text comparing our technique with some previously proposed treatment methods.

One common strategy is to treat sites with the highest likelihoods of ignition (as in the simple scenario we compare to our solutions in Fig. 8). However, this strategy does not optimally reduce the risk of spread of escaped fires nor does it address the uncertainty of determining the sites with the highest fire ignition potential. By comparison, our probabilistic fireshed strategy compartmentalizes regions with high ignition potential, thus providing a hedge against the possibility of fires escaping to spread elsewhere.

Several other fuel treatment strategies have used site-specific priority weights. Minas et al. (2015) linked site treatment and deployment resources to minimize the number of sites covered by these activities. Each site was assigned a weight by ignition probability and the value under risk if a fire originating in that site is not contained by the initial response. Rachmawati et al. (2015) focused on rapid fuel accumulation after treatment and used site-based combinations of vegetation type and age since fire to find an optimal multi-period sequence of fuel treatments. Wei (2012) applied optimization of fuel treatment at a very small scale (7x7 rows) without embedding a fire simulation model but examined the geometry of the treated areas. Finney et al. (2004; 2007) proposed the assessment of fuel treatments by dividing the landscape into rectangular strips oriented normal to the predominant wind directions. Then, fire growth was simulated, starting with the strip farthest upwind, to identify key fire spread routes and their intersections with the potential treatment areas. The process was repeated after moving each strip in the direction of the wind to impact downwind travel routes and subsequent treatment areas. This method finds fuel treatment configurations for a set of likely fire spread routes but overlooks the combinatorial aspects when allocating multiple treatments under a limited budget. Another network-based approach aimed to minimize the connectivity between sites with high fuel loads (Matsypura et al. 2019). Pais et al. (2021) used a network flow model to control the spatial contiguity of the treated area and prioritized treatments using a site-based fire risk metric (the Downstream Protection Value, DPV). The DPV metric assigns treatment priority ranks to sites by modeling fire propagation through a forest landscape as a tree graph and accounts for the potential of a fire ignited at a given site to burn other sites. In contrast, our model makes decisions using the fire spread probabilities between pairs of locations, which enables control based on the presence of possible fire spread paths between these locations. 

Reviewer 2 has indicated that the length of the methods should be reduced, but expects additions around describing the study area and Burn-P3 parameterization.

We wanted to note that the Methods section includes the formulation and description of the Critical Node Detection model. The new model formulation is a result on its own but, in keeping with tradition, is presented in the Methods section. This explains the larger-than-normal section size. We believe that the CND model requires a detailed description to understand its principles, and so keeping the larger section size felt justified. However, the fire behaviour simulation model was already published previously in Reimer at al. (2019), so we have only included a brief summary of this model in the main text and moved the description of the its parameters to Supplement S1.

Reviewer 1 has offered some specific suggestions for improvements in the methods section and the authors should consider moving some text to supplementary materials as suggested by Reviewer 2. Though I found the methods to be a lot to wade through, much of it is necessary in my mind.

We agree that much of the descriptive material in the Methods section is necessary to understand how the model works. To reduce the section size, we have moved a portion of text describing the Burn-P3 model and a table with the model parameters to Supplement S1.

2. We note that Figure 3 in your submission contain map images which may be copyrighted. <…>We require you to either (1) present written permission from the copyright holder to publish these figures specifically under the CC BY 4.0 license, or (2) remove the figures from your submission:

Figure 3 is our own design. While it resembles the figure from Reimer et al. (2019), it was created in GIS from scratch using Burn-P3 inputs and data layers in the public domain, and therefore does not require permission to publish.

Reviewer’s 1 comments:

Reviewer #1: General comments:

First, I suggest you revisit the abstract to make sure it aligns with what you accomplished. Specific notes follow.

We have edited the abstract to make it more succinct.

Fire behavior is a general term that includes spread but also intensity, duration, and type. In most cases, I think it would be clearer to say something like burn probability or spread likelihood instead in this paper.

We have used fire spread in the text to avoid confusion.

Does the problem 3 formulation with time add value to your analysis? As you point out, you do not actually remodel the fire spread probabilities. My personal view is that there is little value planning far in advance in fire-prone landscapes because stochastic wildfire activity will alter the priorities for subsequent periods more than the treatments.

We agree with the reviewer about there being little value to long-term planning in fire-prone landscapes and perceive our approach as a short-term planning tool. Nevertheless, even short-term treatments are typically implemented in a stepwise fashion over a defined time period. The idea behind the problem 3 formulation is to account for the cumulative nature of multi-step planning per se; that is, the actions taken in the first step have the most impact on the system and may thus affect the actions taken in the subsequent steps. The primary difference between the problem 2 and 3 formulations is that the problem 3 formulation allocates the treatments with the greatest impact on fire spread probabilities first, followed by less impactful treatments. We have edited the text to make this point clearer.

If you are interested, other researchers have examined the effects of uncertainty in fire occurrence using two stage models solved with backwards induction:

Konoshima M, Montgomery CA, Albers HJ, Arthur JL (2008) Spatial-endogenous fire risk and efficient fuel management and timber harvest. Land Economics 84(3), 449-468.

Konoshima M, Albers HJ, Montgomery CA, Arthur JL (2010) Optimal spatial patterns of fuel management and timber harvest with fire risk. Canadian Journal of Forest Research 40, 95-108.

These should probably be cited in the discussion when discussing multiple planning periods. 

The suggested references and a brief section of text referring to two-stage models have been added to the Introduction section:

Konoshima et al. (2008, 2010) integrated a fire simulation model into a two-period stochastic dynamic model to find spatial allocations of timber harvest and fuel management in the face of spatially endogenous fire risk. Their approach used a fire simulation model to enumerate all possible fire occurrence patterns in all plausible treatment decisions and considered the trade-offs between fire risk, timber harvest value and fuel treatment cost.

I’m surprised you advocate for problem 3 formulation in the discussion given that you say it produced similar results as problem 2 but with added complexity. Why?

Problems 2 and 3 produced similar spatial results but different allocation of treatments in time. Problem 3 helps find the optimal sequence of treatments (e.g., prescribed burns), whereas problem 2 does not address the issue of optimal timing because it solves only one CND network for an entire planning horizon. In a sense, problem 3 is more realistic because it provides guidance to forest managers about a stepwise implementation of the fuel reduction strategy.

If you had more space, I would recommend you compare your method to something simple and widely used, such as Finney’s treatment optimization method or specific patterns of fuel treatments. When you start to describe the contexts that your problem 3 can be applied in (small landscapes, short durations), I start to wonder if you really diverge from a plan that interrupts the major spread paths of an anticipated problem fire scenario.

Finney MA (2004) Chapter 9, Landscape fire simulation and fuel treatment optimization. In: J.L. Hayes, A.A. Ager, J.R. Barbour, (tech. eds). Methods for integrated modeling of landscape change: Interior Northwest Landscape Analysis System. PNW-GTR-610. p 117-131.

Finney MA (2001) Design of regular landscape fuel treatment patterns for modifying fire growth and behavior. Forest Science 47(2), 219-228.

The only limitation for problem 3 is high numerical complexity. The model can be applied to larger landscapes at a coarser spatial resolution. We have added a brief discussion comparing our approach with other methods – see our reply to the first comment from the Academic Editor.

Specific comments:

Lines 40-42: It is fair to point out that resources to mitigate wildfire risk are limited, and it is therefore important to prioritize, but there are many tools to assist forest managers in planning fuel treatments. I suggest dropping the focus on limited tools here. Whether the tools are used by managers is another issue.

The text discussing limited tools has been dropped following the reviewer’s suggestion.

Line 42: I suggest replacing “interdiction” with common language since you felt the need to define it in the paper.

We have dropped the mention of interdiction – critical node detection is already a good description of the approach.

Line 43: Did you really cover consequences in this work? I didn’t see an effects analysis here.

Yes – essentially, this is what the CND formulation in problems 1-3 does.

Lines 44-45: How about “We used simulation modeling to estimate the likelihood of fire spread between forest network nodes and we…”

The text has been edited as suggested by the reviewer.

Line 46: fuel treatment ALLOCATION problem

This text fragment was deleted.

Lines 64-68: Yes. Costs tend to rise with fire sizes, but rising suppression costs are often attributed to the expansion of human values into wildlands. It would be less controversial if you focused this statement only on the challenge of suppressing fires in rugged landscapes.

We have edited the text to focus the statement on managing fires in rugged landscapes.

Lines 69-70: I think it is appropriate to at least mention that not all forestry and fire scientists agree that fuel treatments will reduce fire spread. Your Agee reference on shaded fuelbreaks includes discussion of this. Agee and Skinner (2005) and Reinhardt et al. (2008) argue strongly that most fuel treatments are, or should be, aimed at reducing fire intensity and severity.

We agree with the reviewer that the focus should not entirely be placed on spread and have added references to fire severity. Although limiting fire spread defines the proposed strategies, we should have better acknowledged the benefits of fuel treatments in reducing fire intensity and severity with respect to the intended purpose of the fuel treatments. In addition to the probability of fire spread, our approach could incorporate other fire behaviour parameters (such as fire intensity) in conjunction with spread as long as such data could be generated with the fire simulation models. We added some explanatory text about this to the Discussion. 

Lines 70-72: Is the saving costs statement supported by these citations? It was a modeling study, but the Thompson et al. (2013) ‘Quantifying the Potential Impacts of Fuel Treatments on Wildfire Suppression Costs’ article provides the clearest estimates of how fuel treatments could reduce costs via their effects on fire sizes.

We have added text referring to Thompson et al. (2013) as the reviewer suggested.

Lines 76-77: Again, I think it is appropriate to acknowledge that some of these models were aimed at reducing the severity of effects instead of large fire spread.

We have added text acknowledging that some models were also designed to reduce the severity of future fires in the landscape. 

Lines 115-116: I would drop the subscripts here and save them for the methods.

Done.

Lines 127-130: Again, I think you should temper this statement to make it clear that it is more of an assumption supported by rules of thumb than a clear conclusion of the research. It is also prudent to acknowledge that fuel treatments do not generally achieve 100% protection, especially in the case of extreme weather. The Kalies and Kent (2016) review on fuel treatment effectiveness may be worth mentioning here.

The statement has been edited to make clear that this assumption is a simplification and we acknowledge that fuel treatments are not usually 100% effective. A citation to Kalies and Kent (2016) has been added to the text.

Line 134: You already introduced the CND abbreviation.

Dropped.

Line 156-157: Would it not be simpler to introduce the model as area limited since you don’t account for variable costs? I see the future value of accounting for this, but it adds slight confusion to the paper. For example, you describe the model as having an upper bound for Rx fire area in the abstract and introduction.

We have replaced the budget limit with a treatment area limit as suggested by the reviewer.

Lines 183-185: And fire suppression?

Edited.

Line 197: “depict well” to “represent”?

Edited

Line 234: “Consecutively” or “consequently”?

Consequently.

Lines 313-314: I do not think it is a good idea to use T and t for both steps in problem 2 and time periods in problem 3. I suggest changing a different letter for the steps in problem 2 to avoid confusion.

We have denoted time periods and the full timespan in problem 3 as t’ and T’, respectively, while keeping t (a planning step) and T (the set of all planning steps) for problem 2. We opted to use the same letter to highlight the analogies between the problem 2 and problem 3 formulations.

Lines 405-407: Rephrase for clarity.

The sentence was deleted.

Lines 437-442: You should clarify exactly how the information you mention was used. Were treatments limited to a particular vegetation type? Did you use the ignition probability, but not spread probability components of Burn P3? Is this later what you refer to as prioritizing on ignitions?

No – this scenario only used a binary map of flammable / non-flammable land cover types (like vegetation and natural fire barriers) and the flammable land classes were expected to support the spread of fires. Comparatively, the scenarios using fire behaviour information utilized fire spread probabilities pij calculated with the fire simulation model.

The only scenario that used ignition probabilities instead of pij values was the solution that allocated treatments to minimize the ignition probability in Fig 8. We have edited the text to make this clearer.

Lines 443-452: This is where it would help to know the difference between T steps and T time periods.

We have changed the notation for time periods in the problem 3 formulation to t’ (and the notation for the full timespan to T’) where appropriate. 

Lines 464-474: As noted in my general comments, this is an important enough change in methods that you should clarify which results it applies to (all?) and describe it fully in the methods section instead of supplementary material referenced from the results.

The calculation of the wij values is not related to the CND model per se – these values were used only to map the fire spread hotspots between multiple pairs of locations. Direct mapping of pij arcs makes the map cluttered and difficult to read. The new mapping procedure of fire spread probabilities is a novelty on its own and could help better understand the fire spread patterns in complex landscapes. The full description of the wij mapping procedure is beyond the scope of the current study – this is the focus of another manuscript. The text provides only basic description germane to understanding the fire spread probability maps. The text has been edited to make this aspect clearer.

Lines 488-489: “ignoring spatial contiguity rules”? or “ignoring the simulated connectivity measures”?

Ignoring spatial contiguity constraints for prescribed treatments – the sentence has been edited.

Line 494: Suggest changing fire behavior to fire spread.

Edited.

Lines 520-525: Why is this scenario suddenly popping up in the results? This should be introduced earlier with justification for what it tells you. Prescribed fires likely reduce ignition risk for a short period after treatment, but this will not last long as fuels reaccumulate. Reducing ignitions with rules and enforcement may require different methods in some landscapes.

We have introduced a scenario minimizing ignition probabilities (by preferentially treating the sites with the highest probabilities) in the Methods section, after the description of the scenario based on land cover information only. In our case, we have only done a basic comparison of methods with the same treatment area. Implementing a practical enforcement scenario that reduces the probability of ignitions would require adapting both the CND and ignition-minimizing scenarios to the current practical standards and is considered as a theme for another manuscript.

Lines 579-582: I’m confused about what scenarios are being compared here.

These sentences were dropped to avoid confusion.

Lines 598-611: I’m wondering how much the small/large fire size tradeoff that is important at this site pertains to the use of probability vs. binary fireshed weighting versus the specific pattern of fire sizes and occurrence on your landscape? What do you think you would find on a landscape with high probability of fire spread from less frequent but large fires?

Given that the model behaviour depends on the spatial configuration of fire spread patterns in the landscape, this question may best be answered by testing the CND model on real landscapes with high probabilities of large fires (e.g., with large amounts of old-growth forest and significant fuel accumulation of after decades of active fire suppression). This could be the focus of future work. To target a particular range of fire sizes, the pij value for each pair of locations i and j could be adjusted in the objective equation by a user-defined coefficient based on the distance between i and j. This would make the approach adaptable to other fire regime conditions and management objectives. The extent of this adaptability to different fire regimes will be examined in further studies.

Lines 615-619: Did the approximation you made to get at spread paths within fires really “address” the problem? I don’t have a brilliant solution to do better without complicating the simulation. I would be tempering my language here to reflect that some approximations were made to prototype a model framework.

We did not use approximation to calculate the pij probabilities of fire spread between pairs of locations. As shown in Supplement 1, the pij calculations used raw fire simulation model outputs (i.e., the ignition points and perimeters of individual simulated fires). A pij value only defines the probability that a fire ignited in location i will spread to location j, but does not specify how exactly the fire could spread from i to j; in short, the CND model does not require exact specification of the fire spread paths between i and j. 

In our study, we have used the ignition points and perimeters of the simulated fires, but did not track specific (and possibly dynamic) fire spread paths within individual fires. Tracking daily or hourly fire spread within individual fire perimeters could potentially refine the pij values, particularly for long spread distances, but would require a more sophisticated fire simulation model that can output the expansion of individual fires on an hourly or daily basis. This would require updating the Burn-P3 simulation model and could be a topic for future research.

Line 625: With a shortest path approximation…

No, there is no shortest path approximation in the calculation of the pij values. The pij values were calculated directly from raw fire simulation model outputs (i.e., simulated fire perimeters and fire ignition points, as described in Supplement S2). The CND model only needs to know the probability that a fire ignited in location i will spread to location j and does not require specification of a spread path from i to j. Accounting for the presence of a path connecting a pair of locations i and j is handled by the CND model constraints (3) and (4). 

Note that we did use a shortest path approximation to calculate the wij values, but this metric is not used in the CND model and was utilized only for mapping the fire spread patterns.

Figure 3: You should probably include a scale bar and north arrow in the study site panel.

We have added a scale bar and north arrow to Figure 3.

Reviewer’s 2 comments: 

The Introduction section is well written and provides a generally good overview of the works that investigated this topic. I only have a remark. The authors limit the Introduction section focusing on previous works carried out in forest areas and using prescribed burnings, while the applicability of the approach they propose could be expanded also to semiarid or rural areas, as well as to fuel management strategies different than prescribed fires (see for instance, among others, Archibald et al. 2005, https://esajournals.onlinelibrary.wiley.com/doi/abs/10.1890/03-5210; Davies et al. 2015, http://dx.doi.org/10.1071/WF15055; Salis et al. 2018, https://www.sciencedirect.com/science/article/pii/S0301479718301191; Prichard et al. 2020 https://pubmed.ncbi.nlm.nih.gov/32086976/ doi:10.3390/f6062148). This is a shortcoming that can be improved.

Yes, the method can be applied in semi-arid and rural areas if spatial fire simulation models capable of producing realistic fire ignition and spread patterns are available. We felt that this statement fits better in Discussion and have added a brief version of the text suggested by the reviewer at the end of the Discussion section.

The section related to Material and Methods is on the whole complete but suffers from the excessive length of the text (from L122 to L384, to L452 including the Case Study Description). Even if manuscripts published in Plos One can be any length, there is need to reduce this part and omit some redundant sentences. Some specific points to improve this section will be provided in later rows.

We wanted to note that the methods section includes the formulation and description of the new Critical Node Detection model. The model formulation represents a new result on its own but is traditionally presented in the Methods section. The model formulation also required detailed explanations to understand its principles. Nevertheless, we have edited the Methods section, reducing textual descriptions wherever possible, eliminating redundancies and moving a portion of the Burn-P3 fire model description to Supplement S1.

The Discussion section needs to be improved, as the comparison between the results and approach presented in this work are not compared with those obtained in other similar works.

We have also added a short discussion describing the other treatment strategies – see our reply to the first comment from the Academic Editor.

Specific comments:

Introduction: The Introduction section seem too much focused on works carried out in forest areas and application of prescribed burnings, while the approach proposed in this study could be expanded also to semiarid or rural areas, as well as to fuel management strategies different than prescribed fires (see for instance, among others, Archibald et al. 2005, https://esajournals.onlinelibrary.wiley.com/doi/abs/10.1890/03-5210; Davies et al. 2015, http://dx.doi.org/10.1071/WF15055; Salis et al. 2018, https://www.sciencedirect.com/science/article/pii/S0301479718301191; Prichard et al. 2020 https://pubmed.ncbi.nlm.nih.gov/32086976/ doi:10.3390/f6062148).

We have followed the reviewer’s suggestion and added text referring to potential problem applications in semiarid and rural areas, and for fuel management options other than prescribed fires (as long as fire simulation models capable of generating realistic fire ignition and spread patterns are available).

L386-389: Please include the size of the study area, as well as the total size of the modeling domain (that is, including the buffer area).

The study area corresponds to the size of the modelling domain (approximately 834 km2). The network included both a core area and a buffer area. We have noted the size of the study area in the text.

L399-402: Considering that no information is provided on crown fire and spot fire settings, I suppose the authors applied Burn-P3 model to simulate surface fire spread. In case the authors simulated crown fires and spot fires, I would recommend including more details on this.

We’ve added the following text explaining how Burn-P3 simulates fires: 

Burn-P3 fully implements the crown fire scheme of the Canadian Fire Behaviour Prediction System (FBP), modelling surface fires as well as the transition to crown fires (and the rate of crown fire spread itself). 

We have also provided a short Burn-P3 summary in the online Supplement S1 along with the key model parameters: 

The critical weather conditions under which the transition from surface to crown fire occurs are dependent on the fuel type. While spot fires are not discretely modelled within the FBP System, the empirical rate of spread equations are based on wildfire observation data for high-intensity crown fires, thus effectively incorporating the role of spot fires and ember transport into the rate of spread models.

L408-410: Please include a table, in the Supplementary data, to summarize the main input data used for fire simulations (e.g.: weather scenarios tested, wind directions and speed, fire spread durations, etc.).

We have added a short summary and a table summarizing the main inputs for Burn-P3 simulations to online Supplement S1.

L435-439: Problems 1-4 related to the critical nodes detection (CND) were introduced in the first equations, several pages before this part. I would recommend helping readers and clarifying that these problems refer to the first equations and the detection of critical nodes.

We have edited the text naming problems 1-3.

L533: Starting a sentence with “Recall that” might be inappropriate, please check

Removed.

L613-666: The Discussion section summarizes relatively well the principles and generalizations from results as well as the significance of results. On the other hand, it does not discuss the results and methods presented in this work in relation to those of others. This is a limitation, so I recommend improving the Discussion in this sense.

We have added text relating the presented method to other planning fuel treatment methods to Discussion section – see our reply to the first comment from the Academic Editor. To avoid repetitions and keep the size of the manuscript reasonable we have compared our approach with the most common methods that use site-based fire hazard metrics.

Figures 3-4: Please include the scale bar.

We have added scale bars to Figures 3 and 4.

---

## [Decision Letter · Decision Letter 1]

1 Sep 2021

PONE-D-21-15640R1

DETECTING CRITICAL NODES IN FOREST LANDSCAPE NETWORKS TO REDUCE WILDFIRE SPREAD

PLOS ONE

Dear Dr. Yemshanov,

Thank you for submitting your manuscript to PLOS ONE. After careful consideration, we feel that it has merit but does not fully meet PLOS ONE’s publication criteria as it currently stands. Therefore, we invite you to submit a revised version of the manuscript that addresses the points raised during the review process.

Reviewer 1 has expressed that the manuscript can be accepted pending some minor revisions. 

Reviewer 2 was unavailable to review the revision, so I reviewed the authors’ revisions based on the reviewer’s comments and found that all issues have been adequately addressed. 

Lines 326-328, “In general, long-term planning in fire-prone landscapes has little utility because stochastic wildfire activity may override the long-term treatment plans.” is very poor phrasing and inaccurate. Long term planning is a hallmark of forest management generally and there is good work being undertaken at a range of temporal scales that specifically address stochasticity in fire regimes. There is no need for this baseless claim and it should be removed.

We look forward to receiving your revised manuscript.

Kind regards,

Paul Pickell, Ph.D.

Academic Editor

PLOS ONE

Journal Requirements:

Reviewers' comments:

Reviewer's Responses to Questions

**Comments to the Author**

1. If the authors have adequately addressed your comments raised in a previous round of review and you feel that this manuscript is now acceptable for publication, you may indicate that here to bypass the “Comments to the Author” section, enter your conflict of interest statement in the “Confidential to Editor” section, and submit your "Accept" recommendation.

Reviewer #1: (No Response)

2. Is the manuscript technically sound, and do the data support the conclusions?

Reviewer #1: Yes

3. Has the statistical analysis been performed appropriately and rigorously? 

Reviewer #1: Yes

4. Have the authors made all data underlying the findings in their manuscript fully available?

Reviewer #1: Yes

5. Is the manuscript presented in an intelligible fashion and written in standard English?

Reviewer #1: Yes

6. Review Comments to the Author

Reviewer #1: I am pleased with the revisions. Thank you for clarifying your methods and for explaining where you think the multi-year planning scenario adds value. I still think the method of defining spread probabilities between nodes using the shortest paths from ignition points to perimeters should be framed as an approximation, but this is not central to your work, and the detailed spread paths may not matter much if the “nodes” are large.

A few minor writing suggestions:

L49-51: I would edit this sentence as: “Our results provide new insights into cost-effective planning to mitigate wildfire risk in forest landscapes. The approach should be applicable to other ecosystems with frequent wildfires.”

L64: Drop “in places”?

L78: “fuel treatments” instead of “fuel treatment measures”

L132: Drop “area”

L430-436: Are these sentences necessary after you simplified to an area/node count limit?

7. PLOS authors have the option to publish the peer review history of their article (what does this mean?). If published, this will include your full peer review and any attached files.

Reviewer #1: No

---

## [Author Response · Author response to Decision Letter 1]

3 Sep 2021

Manuscript PONE-D-21-15640R1 “DETECTING CRITICAL NODES IN FOREST LANDSCAPE NETWORKS TO REDUCE WILDFIRE SPREAD” – A Reply to Reviewers’ Comments:

Academic Editor:

Lines 326-328, “In general, long-term planning in fire-prone landscapes has little utility because stochastic wildfire activity may override the long-term treatment plans.” is very poor phrasing and inaccurate. Long term planning is a hallmark of forest management generally and there is good work being undertaken at a range of temporal scales that specifically address stochasticity in fire regimes. There is no need for this baseless claim and it should be removed.

We deleted this text as suggested by the Editor.

Reviewer #1: 

I am pleased with the revisions. Thank you for clarifying your methods and for explaining where you think the multi-year planning scenario adds value. I still think the method of defining spread probabilities between nodes using the shortest paths from ignition points to perimeters should be framed as an approximation, but this is not central to your work, and the detailed spread paths may not matter much if the “nodes” are large.

We want to clarify that the calculation of fire spread probabilities pij between pairs of nodes (which were used in our optimization model) did not involve a shortest path approximation. Recall that the spread probability value pij depicts the likelihood that a fire ignited in node i spreads to node j without indication of how (i.e., by what path) the fire might spread from i to j. Information about the ignition locations came from the geographic coordinates for individual fires simulated by the Burn-P3 model, and then we used the perimeters of the simulated fires to identify the locations j to which a particular fire ignited in i could spread.

The only place where we utilized a shortest path approximation was in the calculation of the spread probabilities wij between adjacent nodes for visualizing the fire spread patterns. For each simulated fire, we used the shortest path approximation to project the possible fire spread path (with the probability pij) between locations i and j over the network of arcs connecting the adjacent nodes. This map only served to illustrate the fire spread patterns and was not used in optimization modelling. We have edited two sections of the main text, “Calculating the fire spread probabilities pij” and “Mapping the fire spread probabilities”, as well as Supplement S4 to make this aspect clearer.

A few minor writing suggestions:

L49-51: I would edit this sentence as: “Our results provide new insights into cost-effective planning to mitigate wildfire risk in forest landscapes. The approach should be applicable to other ecosystems with frequent wildfires.”

Edited as the reviewer suggested.

L64: Drop “in places”?

Dropped.

L78: “fuel treatments” instead of “fuel treatment measures”

Corrected.

L132: Drop “area”

Dropped.

L430-436: Are these sentences necessary after you simplified to an area/node count limit?

These sentences provide background about why we simplified the budget calculations to an area/node count limit and so should stay in the text. Note that the budget constraint [2] would require a variable cost component if the site treatments were solely managed by ground crews, in which case the total treatment cost would depend on the time required to access the treatment sites (which could be a function of complex terrain and proximity to roads).

---

## [Editor Report · Decision Letter 2]

17 Sep 2021

DETECTING CRITICAL NODES IN FOREST LANDSCAPE NETWORKS TO REDUCE WILDFIRE SPREAD

PONE-D-21-15640R2

Dear Dr. Yemshanov,

We’re pleased to inform you that your manuscript has been judged scientifically suitable for publication and will be formally accepted for publication once it meets all outstanding technical requirements.

Kind regards,

Paul Pickell, Ph.D.

Academic Editor

PLOS ONE
---

## [Editor Report · Acceptance letter]

21 Sep 2021

PONE-D-21-15640R2 

DETECTING CRITICAL NODES IN FOREST LANDSCAPE NETWORKS TO REDUCE WILDFIRE SPREAD 

Dear Dr. Yemshanov:

I'm pleased to inform you that your manuscript has been deemed suitable for publication in PLOS ONE. Congratulations! Your manuscript is now with our production department. 

Kind regards, 

on behalf of

Dr. Paul Pickell 

Academic Editor

PLOS ONE